# Multimodal analysis of cfDNA methylomes for early detecting esophageal squamous cell carcinoma and precancerous lesions

Jiaqi Liu [1,2,6], Lijun Dai[3,6], Qiang Wang[4,6], Chenghao Li[3,6], Zhichao Liu [5], Tongyang Gong [1], Hengyi Xu[1], Ziqi Jia[2], Wanyuan Sun[1], Xinyu Wang[3], Minyi Lu[1], Tongxuan Shang[2], Ning Zhao[1], Jiahui Cai[1], Zhigang Li [5], Hongyan Chen [1] ✉, Jianzhong Su [3] ✉ & Zhihua Liu [1] ✉

Detecting early-stage esophageal squamous cell carcinoma (ESCC) and precancerous lesions is critical for improving survival. Here, we conduct whole-genome bisulfite sequencing (WGBS) on 460 cfDNA samples from patients with non-metastatic ESCC or precancerous lesions and matched healthy controls. We develop an expanded multimodal analysis (EMMA) framework to simultaneously identify cfDNA methylation, copy number variants (CNVs), and fragmentation markers in cfDNA WGBS data. cfDNA methylation markers are the earliest and most sensitive, detectable in 70% of ESCCs and 50% of precancerous lesions, and associated with molecular subtypes and tumor microenvironments. CNVs and fragmentation features show high specificity but are linked to late-stage disease. EMMA significantly improves detection rates, increasing AUCs from 0.90 to 0.99, and detects 87% of ESCCs and 62% of precancerous lesions with >95% specificity in validation cohorts. Our findings demonstrate the potential of multimodal analysis of cfDNA methylome for early detection and monitoring of molecular characteristics in ESCC.

Esophageal cancer (EC) is one of the most prevalent gastrointestinal malignancies and the sixth leading cause of cancer-related mortality worldwide[1]. Esophageal squamous cell carcinoma (ESCC), the predominant histological subtype of EC, accounts for approximately 88% of new EC cases, with the majority occurring in Eastern and Central Asia[2]. ESCC exhibits a dismal prognosis characterized by one of the lowest 5-year overall survival rates (18.5% and 36.9% in the US and China, respectively)[3], which is largely attributed to late-stage diagnosis[4]. In contrast, early-stage ESCC, such as intramucosal ESCC,

and precursor lesions like intraepithelial neoplasia (IEN), can achieve nearly 100% five-year disease-specific survival rates through endoscopic *en bloc* resection, obviating the need for systematic treatment[5–7]. Therefore, early detection is critical for enhancing the survival and quality of life for ESCC patients.

The gold standard for diagnosing ESCC and its precursor lesions remains endoscopy with iodine staining[8]. However, the widespread adoption of endoscopic screening faces challenges, including low compliance and the substantial cost of conducting endoscopic

[1]State Key Laboratory of Molecular Oncology, National Cancer Center/National Clinical Research Center for Cancer/Cancer Hospital, Chinese Academy of Medical Sciences and Peking Union Medical College, 100021 Beijing, China. [2]Department of Breast Surgical Oncology, National Cancer Center/National Clinical Research Center for Cancer/Cancer Hospital, Chinese Academy of Medical Sciences and Peking Union Medical College, 100021 Beijing, China. [3]Oujiang Laboratory (Zhejiang Lab for Regenerative Medicine, Vision and Brain Health), Eye Hospital, Wenzhou Medical University, Wenzhou 325027, China. [4]Department of Anesthesiology, National Cancer Center/National Clinical Research Center for Cancer/Cancer Hospital, Chinese Academy of Medical Sciences and Peking Union Medical College, 100021 Beijing, China. [5]Department of Thoracic Surgery, Shanghai Chest Hospital, Shanghai Jiao Tong University School of Medicine, Shanghai 200030, China. [6]These authors contributed equally: Jiaqi Liu, Lijun Dai, Qiang Wang, Chenghao Li. ✉e-mail: chenhongyan@cicams.ac.cn; sujz@wmu.edu.cn; liuzh@cicams.ac.cn

examinations for millions of eligible individuals in high-risk regions like China[9]. Liquid biopsy methods, capable of detecting circulating tumor DNA (ctDNA) within cell-free DNA (cfDNA) in plasma, offer a promising avenue for non-invasive early cancer detection[10,11]. However, few studies have assessed the utility of liquid biopsy in ESCC diagnosis[12].

Advancements in cfDNA biology, coupled with the exponential growth in data volume, have enabled the detection of tumor-specific alterations with unparalleled precision, encompassing genetic[13], epigenetic[14], and fragmentomic features[15]. In a recent multi-omics analysis of cfDNA within the Circulating Cell-free Genome Atlas (CCGA), whole-genome cfDNA methylation emerged as the most promising signal for cancer detection, outperforming fragmentation features and genetic variants[16]. However, there are still some challenges with methylation-based approaches for detecting ctDNA.

First, the intrinsic value of genetic variants (e.g., copy number variants; CNVs) and fragmentation features (e.g., fragment size) embedded within whole-genome bisulfite sequencing (WGBS) data has been underappreciated[17]. A concurrent analysis of cfDNA methylation markers, genetic variants, and fragmentation features within a single WGBS dataset is needed. Second, given the pivotal roles of both genetic and epigenetic aberrations in the transition from precancerous lesions to ESCC[18-20], the mutual complementarity and combined performance of these multi-omics features remain unclear. Third, the biological significance of cfDNA methylation markers and their utility for subtyping and prognosis are largely unknown.

Here, we conducted WGBS on 460 cfDNA samples from 230 patients with non-metastatic ESCC or precancerous lesions and 230 matched healthy controls (HCs) from multiple centers. To simultaneously examine the cancer-associated differentially methylated regions (DMRs), CNVs, and fragmentation features within cfDNA WGBS data, we developed a comprehensive approach termed Expanded Multi-Modal Analysis (EMMA). This approach allowed us to profile the complementarity, temporal dynamics, and detection efficacy of epigenetic and genetic features within cfDNA for ESCC. Eventually, we determined the biological relevance of optimal cfDNA methylation features in this context.

## Results

### Overview of the EMMA framework in cfDNA whole-genome bisulfite sequencing

To enhance the early detection of ESCC, we developed the EMMA framework based on a "tissue-cfDNA-tissue" strategy (Fig. 1a). Initially, we identified ESCC-derived DMRs and CNVs from paired WGBS and whole-genome sequencing (WGS) data of primary tumors and matched adjacent non-neoplastic tissues of 155 ESCC cases from our previous cohort, the ESCC Genome and Epigenome Atlas (ECGEA)[18]. Subsequently, we examined the ESCC-derived DMRs and CNVs in cfDNA WGBS data. Based on the fact that fragments originating from tumor cells appear to be shorter than fragments from normal cells[21], we further calculated the proportion of short cfDNA fragment sizes as the fragment size ratios (FSRs) in cfDNA WGBS data. Next, we employed random forest-based machine learning frameworks, utilizing DMRs alone, DMRs in combination with CNVs, and all three features (DMRs, CNVs, and FSRs), using a dataset containing 150 ESCC patients and 150 matched HCs. The performance of each diagnostic model was independently assessed in an external ESCC cohort and a precancerous cohort. Additionally, we correlated the optimal DMRs with integrated multi-omics-based molecular subtypes, the tumor microenvironment (TME), survivals, and transcriptomic profiles in the paired ESCC tissue samples.

Our study encompassed a diverse cohort, including 150 untreated patients with ESCC or esophageal high-grade intraepithelial neoplasia (HGIEN, namely stage-0 ESCC) from the Cancer Hospital of the Chinese Academy of Medical Sciences and Peking Union Medical College in Beijing, China (CHCAMS, the discovery/training cohort), 30 untreated ESCC patients from the Shanghai Chest Hospital in Shanghai, China (the external validation cohort), 50 patients with esophageal IEN from CHCAMS (the precancerous validation cohort), and 230 HCs, each age- and gender-matched within their respective cohorts (Fig. 1b; Supplementary Table 1). We collected a median of 2 mL of plasma from each participant ($n = 460$) before any medical intervention. The cfDNA concentrations were consistent in ESCC patients and controls (Supplementary Fig. 1). WGBS was employed to assess the cfDNA methylome of each participant, with the 460 cfDNA samples representing approximately 89% coverage of the reference genome with an average depth of 9.51× (Supplementary Fig. 1).

### Identification and performance of cfDNA methylation markers

We initially identified 41,199 DMRs among the 155 ESCC tissues and their corresponding adjacent non-neoplastic tissues in the ECGEA cohort. To quantify the proportion of ctDNA within the cfDNA samples, we utilized a previously established computational framework[14]. This framework relied on specific DMR patterns associated with ESCC. We calculated 'cfDNA malignant ratios' to estimate ctDNA content accurately within the samples. In cfDNA, we identified 650 DMRs by comparing WGBS data from 150 ESCC patients with 150 HCs from CHCAMS. Subsequently, the cfDNA malignant ratio for each DMR was calculated in every sample. Utilizing a random forest algorithm, we generated prediction models based on cfDNA malignant ratios derived from these DMRs, ranging from 2 to 650, using data from the discovery cohort (Supplementary Fig. 1). Consequently, a specific set of 50 optimal DMRs demonstrated the highest discriminatory power in distinguishing between malignant and benign plasma samples (Methods; Supplementary Fig. 2). The final prediction model, denoted as the ESCC-cfMeth score, was constructed using the cfDNA malignant ratios of these 50 markers (Fig. 2a and Supplementary Fig. 3a), including 40 hypo-DMRs and 10 hyper-DMRs. In terms of performance, the ESCC-cfMeth score achieved an area under the curve (AUC) of 0.90 (95% CI, 0.87−0.94) in the 10-fold cross-validation within the discovery cohort. In the external validation cohort, the AUC was 0.89 (95% CI, 0.81−0.98), and in the precancerous validation cohort, it reached 0.87 (95% CI, 0.80−0.94; Fig. 2b). Using a cutoff threshold of 0.5, the prediction model achieved accuracies of 82.33%, 85.00%, and 78.00% in the discovery, external validation, and precancerous validation cohorts, respectively (Methods; Supplementary Table 2).

The ESCC-cfMeth scores were significantly higher in ESCC patients than HCs in both the training and external validation cohorts. Notably, ESCC-cfMeth scores were also increased in patients with IENs in the precancerous cohort (Fig. 2c). However, the scores did not further increase with the progression of the ESCC stages, implying the scores could reflect the biological characteristics of early-stage disease including precancerous lesions, but cannot be used an indication of tumor progression (Fig. 2d). To gain further mechanistic insights, we annotated the genes of the optimal 50 DMRs. Among the 35 annotated genes located within or proximal to the 50 DMRs (within ±2 kb of the gene body or promoter), 16 genes exhibited significant differences at transcription levels in 155 ESCC tissues compared with their corresponding adjacent tissue samples (Supplementary Data 1). Notably, ZNF132, a tumor suppressor gene playing a key role in ESCC development[22], displayed significant down-regulation with a hypermethylated promoter. Conversely, seven genes with hypomethylation within their gene bodies, including FLT1 and LINC00680, which impact cancer cell invasion[23] and promote ESCC progression as competing endogenous RNA[24], respectively, showed upregulation. Additionally, ID1 was up-regulated with a hypomethylated promoter (Fig. 2e), potentially contributing to ESCC tumorigenesis[25]. To reveal the biological significance of these classical functional genes in early-stage ESCC, we analyzed and validated the expression levels of these genes in 10-pair tissue samples of stage-I ESCC and normal tissues from a published dataset [GSE213565][26]. Similarly, ZNF132 was significantly

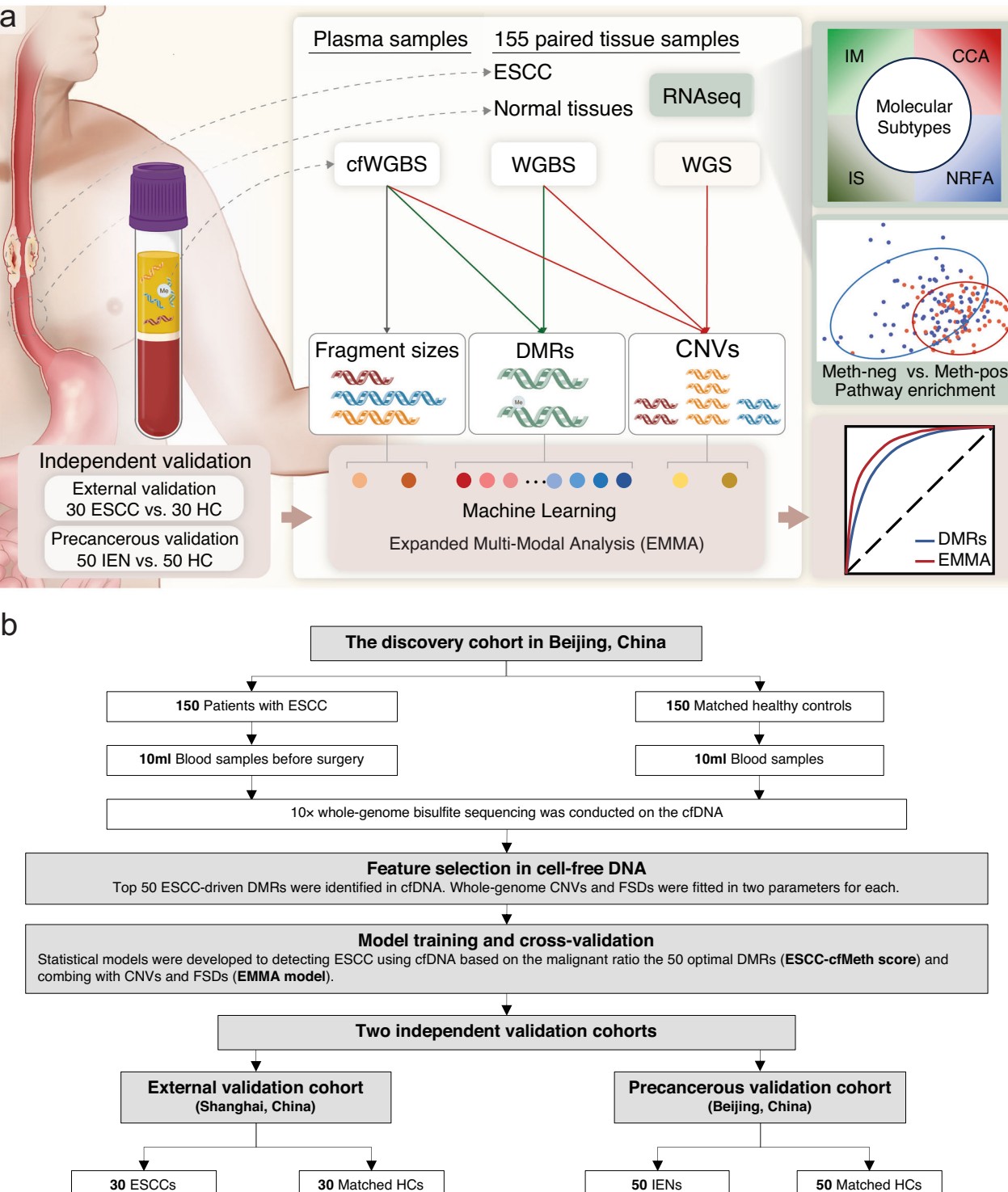

**Fig. 1 | Study design and patient enrollment. a** An approach called 'expanded multimodal analysis' (EMMA) has been developed using machine learning to enhance the detection of ctDNA from cfDNA in plasma samples. This is achieved by comprehensively analyzing cancer-derived differentially methylated regions (DMRs), copy number variants (CNVs), and fragmentation features in the cfDNA whole-genome bisulfite sequencing (cfWGBS) data. The cancer-derived DMRs and CNVs were initially identified from paired WGBS and whole-genome sequencing (WGS) data of primary tumors and matched adjacent non-neoplastic tissues of 155 patients with esophageal squamous cell carcinoma (ESCC). Subsequently, the ESCC-derived DMRs and CNVs were examined in cfWGBS data and further utilized with the proportion of short cfDNA fragment sizes to train the diagnostic models in the discovery cohort. The performance of each diagnostic model was independently assessed in an external ESCC cohort and a precancerous cohort. To unveil

the biological significance of these optimal DMRs, we correlated them with multi-omics-based molecular subtypes and transcriptomic profiles in the paired ESCC tissue samples. **b** The discovery cohort encompassed 150 patients with ESCC or high-grade intraepithelial neoplasia and 150 matched health controls to construct the diagnostic model using different cfDNA features. The performance of each diagnostic model was evaluated independently in an external ESCC cohort and a precancerous cohort. ESCC esophageal squamous cell carcinoma, IEN intraepithelial neoplasia, WGS whole-genome sequencing, WGBS whole-genome bisulfite sequencing, cfWGBS cfDNA WGBS, RNAseq RNA sequencing, HC healthy control, CNV copy number variant, DMR differentially methylated region, IM immune modulation, CCA cell cycle pathway activation, IS immune suppression, NRFA NRF2 oncogenic activation.

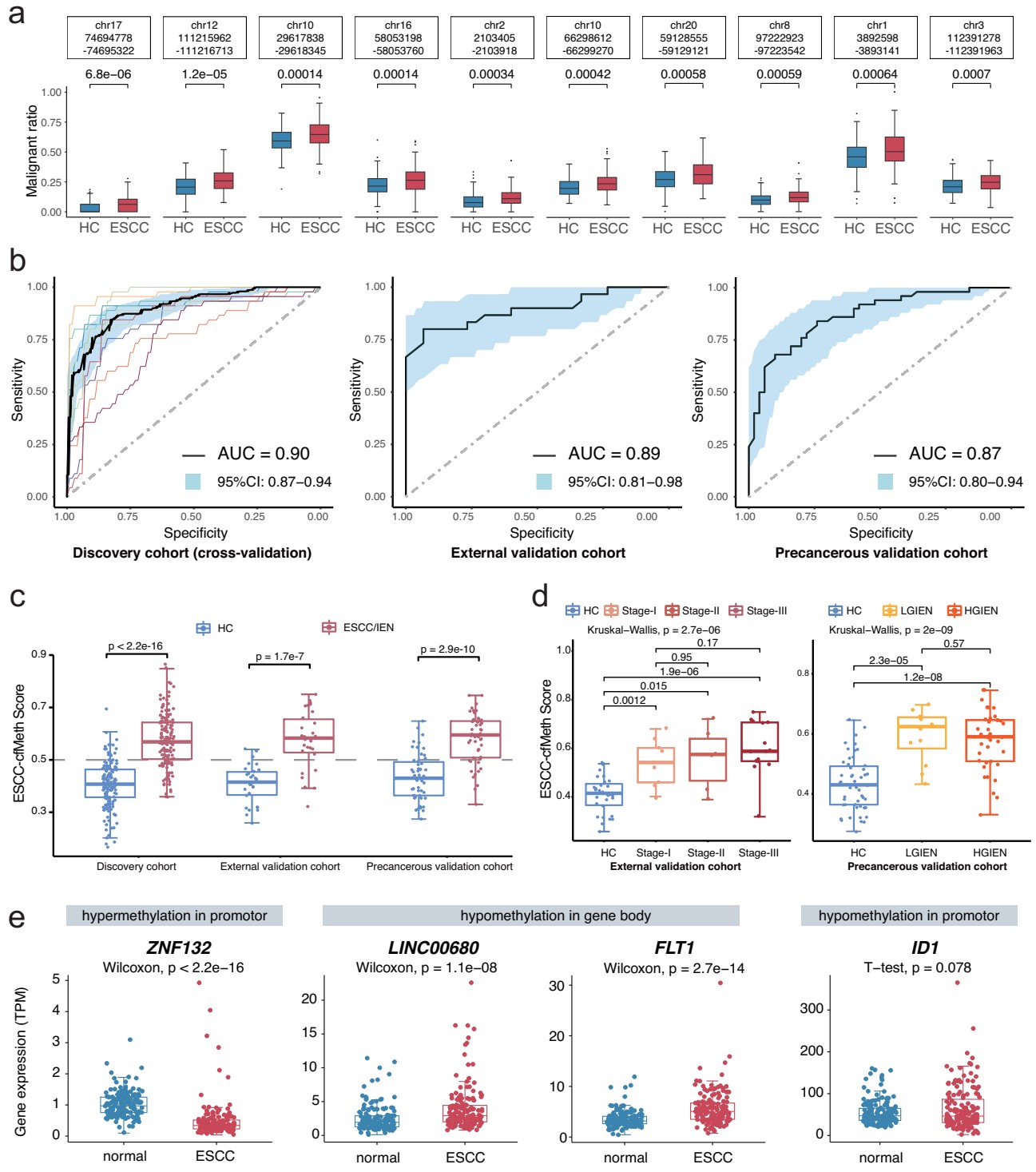

down-regulated and *LINC00680* was significantly up-regulated, which indicates their potential function in early-stage ESCC. However, *FLT1* and *ID1* were also up-regulated but without significance in stage-I ESCC (Supplementary Fig. 3b). Taken together, these findings not only underscore the reliability and interpretability of these cfDNA methylation features for ESCC diagnosis but also indicate their potential functional relevance in ESCC progression.

### Identification of CNVs in cfDNA for ESCC detection

In addition to cfDNA methylation markers, we also focus on the value of cfDNA-derived CNVs for ESCC diagnosis. We initiated this by comparing ESCC tissues with adjacent neoplastic tissues within the ECGEA

cohort using WGS data as the gold standard, establishing a robust foundation for identifying ESCC-derived somatic CNVs. Subsequently, we developed a WGBS-based approach to identify recurrent CNVs in both tissue and cfDNA samples (Fig. 3a). The genome of each sample was partitioned into 1-Mb bins, and a Hidden Markov Model was applied to assess the depth of each bin relative to the baseline established by the software. After CG correlation, log2 ratios were computed for each bin to determine CNV events. Most CNVs identified in the WGS data, including 72.68% of amplifications (1016/1398) and 88.22% of deletions (1153/1307), were successfully identified in paired WGBS data from tissue samples. Patient 002 from the ECGEA cohort serves as an illustrative example, where most amplifications and

**Fig. 2 | Cell-free DNA methylation markers and their detection performance for esophageal squamous cell carcinoma. a** Among the differentially methylated regions (DMRs) identified in esophageal squamous cell carcinoma (ESCC) tissues, 650 DMRs were recalled through an adjusted $p$ value < 0.05 (two-sided Wilcoxon test), favoring DMRs with ESCC average values ($n = 150$) more significant than healthy controls ($n = 150$) in the discovery cohort, as determined by the malignant ratio. The figure shows malignant ratios with the $p$ value of the top ten DMRs as examples. Data are presented as median values with maximums and minimums. **b** The diagnostic performances of the ESCC-cfMeth score were evaluated in the discovery cohort (tenfold cross-validation, the curve of each color indicating one cross-validation), the external validation cohort, and the precancerous validation cohort. The black curves represent the receiver operating characteristic (ROC) curves and the blue areas indicate the 95% confidence intervals (CI). **c** The final prediction model (ESCC-cfMeth score) was constructed using the cfDNA malignant ratios of the optimal 50 markers. The ESCC-cfMeth scores were significantly higher in patients with ESCC and intraepithelial neoplasia (IEN) than the HCs in the discovery cohort and the validation cohorts (two-sided Mann–Whitney U-test, $p < 0.01$). Data are presented as median values with maximums and minimums. **d** Compared to the HCs, the ESCC-cfMeth scores were robustly elevated among ESCCs of different stages (left; $n = 30$, 9, and 15, respectively) and IENs (right; $n = 50$, 12, and 38, respectively). Data are presented as median values with maximums and minimums. **e** Differential expression of functional genes was found within the 50 optimal DMRs in comparison of gene expression between ESCC tissues and paired adjacent non-neoplastic tissues ($n = 155$) using a two-sided Wilcoxon test or $t$ test ($p < 2.2 \times 10^{-16}$ for *ZNF132*, $p = 1.1 \times 10^{-8}$, $2.7 \times 10^{-14}$, and 0.078 for *LINC00680*, *FLT1*, and *ID1*, respectively). Data are presented as median values with maximums and minimums. ESCC esophageal squamous cell carcinoma, IEN intraepithelial neoplasia, LGIEN low-grade IEN, HGIEN high-grade IEN, HC healthy control, DMR differentially methylated region, AUC area under curve, CI confidence interval. Source data are provided as a Source Data file.

deletions identified in WGS data were also observed in the paired WGBS data (Fig. 3b). Subsequently, CNVs detected in both WGS and WGBS data from tissue samples were compared, with only the shared CNVs were retained for further analysis in cfDNA.

To delineate the CNV profile in cfDNA, we employed ichorCNA[27]. In cfDNA, 38.19% of amplifications (388/1016) and 17.17% of deletions (198/1153) identified in WGS and WGBS data from tissue samples were successfully identified. To improve specificity, we focused on regions exhibiting recurrent CNVs in three or more cfDNA samples. In comparison to CNV events in HCs, we found significantly higher CNV event rates in 153 regions in ESCC patients, with 111 amplified regions and 42 deleted regions (Fig. 3c). Within these 153 recurrent CNV regions, 14.67% of ESCC patients in the discovery cohort, 23.33% in the external validation cohort, and 6.00% of IEN patients exhibited CNV events, all of which exceeded the HC groups in their respective cohorts (2.00%, 0%, and 2.00%; $p = 8.3 \times 10^{-5}$, 0.01, and 0.62, respectively; Fisher's exact test). Notably, the incidence of CNV events gradually increased from HGIEN and was positively correlated with cancer stages and grades (Fig. 3d). Collectively, these results highlight the high specificity of CNV events in cfDNA among healthy individuals (ranging from 98.00% to 100%) and their positive correlation with tumor progression and late-stage disease.

There are 583 genes located in the regions of CNV markers in the ESCC cfDNA. Of those genes, 27 and 301 are amplified on chromosomes 3 and 8, respectively, while 120 are deleted on chromosome 4. However, even though our previous whole genome sequencing studies showed that 11q13.3 amplification containing *CCND1* is recurrent in ESCC tissues[19], there was no amplification event was observed on chromosome 11 in the ESCC cfDNA. This could be attributed to the damage inflicted on certain cfDNA fragments by the bisulfite conversion process during WGBS, further attenuating the signals of CNVs in these regions. To reveal the potential function of these genes, we annotated these genes using oncogenes from the ONGene database[28] and analyzed the association between the expression levels of these genes with the prognosis in the tissue samples in our previous ECGEA cohort[18] (Supplementary Data 2). As a result, 41 genes were found to have significant associations with prognosis in ESCC patients, including nineteen genes with amplifications in chromosome 8 and eight genes with deletions in chromosome 4. Some of these genes were reported to be associated with late-stage events in cancer including immune evasion, treatment resistance, and metastasis, such as *FBXO32*[29] within the amplifications in chromosome 8 and *FOXA1*[30] within the amplifications in chromosome 14.

### Fragment size measurement in cfDNA WGBS data

To further reveal the multimodal features of cfDNA, we conducted a comprehensive analysis of cfDNA fragment size profiles in the discovery cohort. Notably, cfDNA samples from both ESCC patients and HCs exhibited peaks at 166 bp. However, a higher proportion of shorter fragments (90–150 bp) was observed in the ESCC groups (Fig. 4a), consistent with previous findings in cfDNA studies related to ESCC[31].

We calculated the FSR by assessing the ratio of short fragments in cfDNA to the human genome. These ratios were determined within 5-Mb bins, yielding 579 FSR features (Fig. 4b, c). No significant difference in the average FSR was found across all bins between ESCC patients and HCs in the discovery cohort. However, we identified 83 bins with significantly elevated FSRs in ESCC patients compared to HCs ($p < 0.05$), indicating position-dependent changes in cfDNA fragmentation. Consequently, compared to matched HCs, the average FSRs within these 83 selected bins were significantly higher in ESCC patients in both the discovery cohort and the external validation cohort ($p = 0.032$ and $2.6 \times 10^{-7}$, respectively). Interestingly, this significant difference was not observed in IEN patients from the precancerous validation cohort ($p = 0.33$; Fig. 4d).

To explore the diagnostic potential of these findings, we developed a predictive model using the FSRs from the 83 selected regions (Methods). However, this model displayed limited discriminatory abilities in both the external validation cohort and the precancerous validation cohort (AUCs = 0.54 and 0.53, respectively; Supplementary Fig. 4). The variable performance of FSR between the ESCCs and IENs suggests a potential correlation between the proportions of short cfDNA fragment sizes and tumor progression.

### Accurate detection of ESCC and precancerous lesions with the combined EMMA model

To improve the diagnostic capability, we integrated genetic and epigenetic features into a combined EMMA model. Notably, the optimal 50 DMRs and 153 CNV regions were predominantly located in different regions of the human genome. Specifically, 74.54% of the optimal DMRs (13769/18980) were located within the 83 selected FSR regions, suggesting that cancer-derived cfDNA was enriched in these regions. In contrast, CNVs were enriched in regions such as 5p and 8q, with less overlap with the FSR regions (Fig. 5a; Supplementary Fig. 5). While there was no significant association between the ESCC-cfMeth score, average FSR, and the CNV events in the discovery cohort (Supplementary Fig. 5), complementary relationships were observed among these features in ESCC patients in the discovery cohort (Fig. 5b).

To reduce the dimensionality of the CNV features in 153 regions and FSRs in the 83 bins, we created two composite parameters for each modal (Fig. 1a). Next, we developed two combined diagnostic models through separate random forest models in the discovery cohort with 10-fold cross-validation: one combining the 50 DMRs with two CNV parameters (DMR *plus* CNV model) and another combining the 50 DMRs with two CNV parameters and two FSR parameters (EMMA model). Both combined models demonstrated significantly improved performance compared to the ESCC-cfMeth score in the discovery cohort as determined by the 10-fold cross-validation (AUC = 0.98, 95%

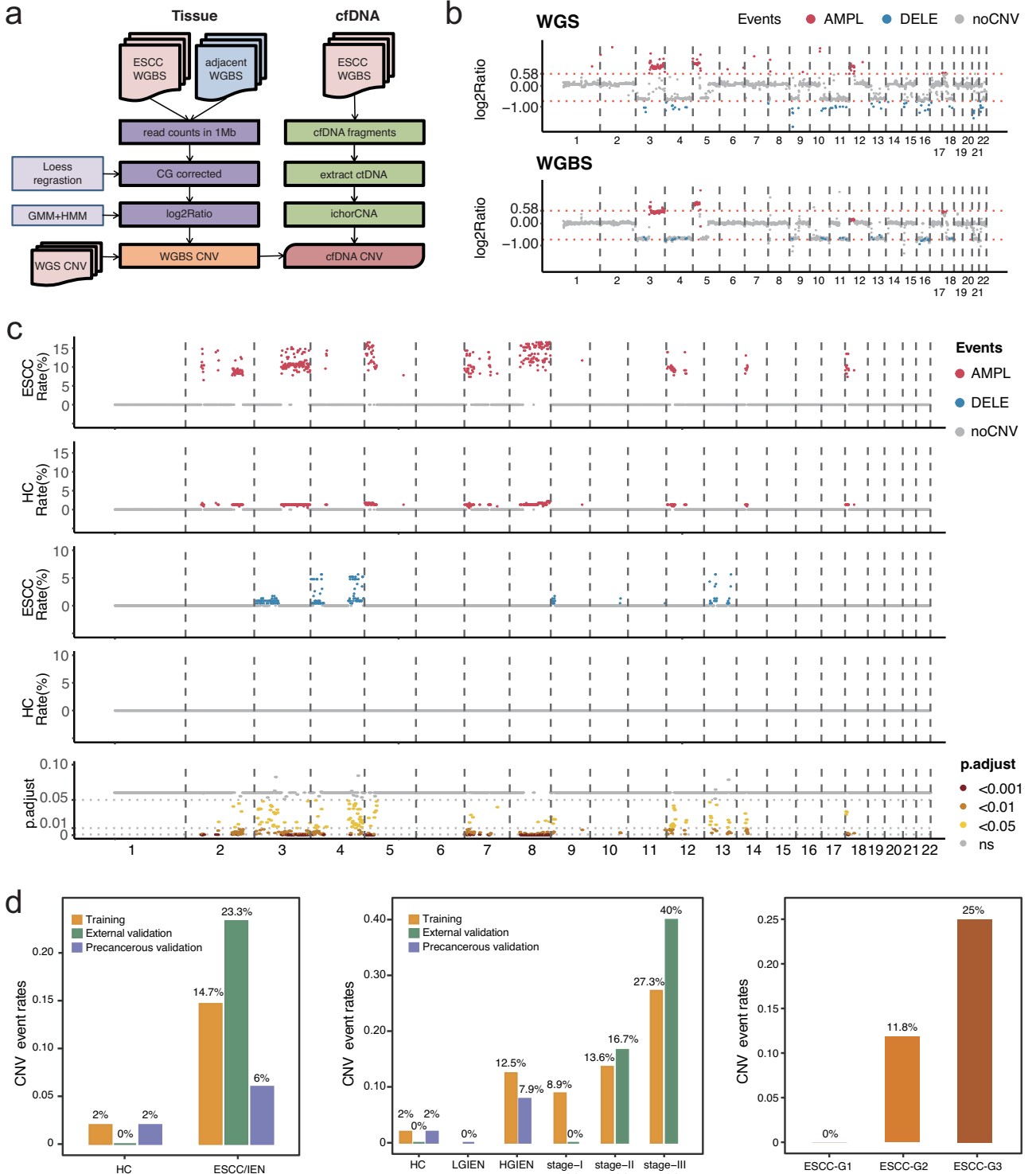

**Fig. 3 | Recalling and analyzing the copy number variant events in cell-free DNA whole-genome bisulfite sequencing data. a** A whole-genome bisulfite sequencing (WGBS)-based approach for recalling recurrent copy number variants (CNVs) in WGBS data from tissues and cfDNA. **b** Take Patient 002 in the ECGEA cohort as an example, the amplifications in chr. 3 and chr.5 and deletions in chr. 3, 4, 9, 10, 11, 13, 16,18, and 21 were identified in whole-genome sequencing data and recalled in paired WGBS data. **c** Compared to the CNV events in 150 healthy controls (HCs), 153 regions had significantly higher CNV event rates in 150 patients with esophageal squamous cell carcinoma (ESCC). The amplifications (red) and deletions (blue) were shown with the corresponding adjusted *p* value (false discovery rate, FDR) of

the difference in ESCCs *vs*. HCs (two-sided *t* test; ns, gray; FDR < 0.05, yellow; FDR < 0.01, orange; FDR < 0.001, dark red). **d** The CNV-positive rates were significantly higher in patients with ESCC and intraepithelial neoplasia (IEN) than the HCs in the discovery cohort and the validation cohorts and positively correlated with the stages and grades. ESCC esophageal squamous cell carcinoma, IEN intraepithelial neoplasia, LGIEN low-grade IEN, HGIEN high-grade IEN, HC healthy control, WGS whole-genome sequencing, WGBS whole-genome bisulfite sequencing, GMM Gaussian Mixture Model, HMM Hidden Markov Model, CNV copy number variant, AMPL amplification, DELE deletion, ns no significance, G grade. Source data are provided as a Source Data file.

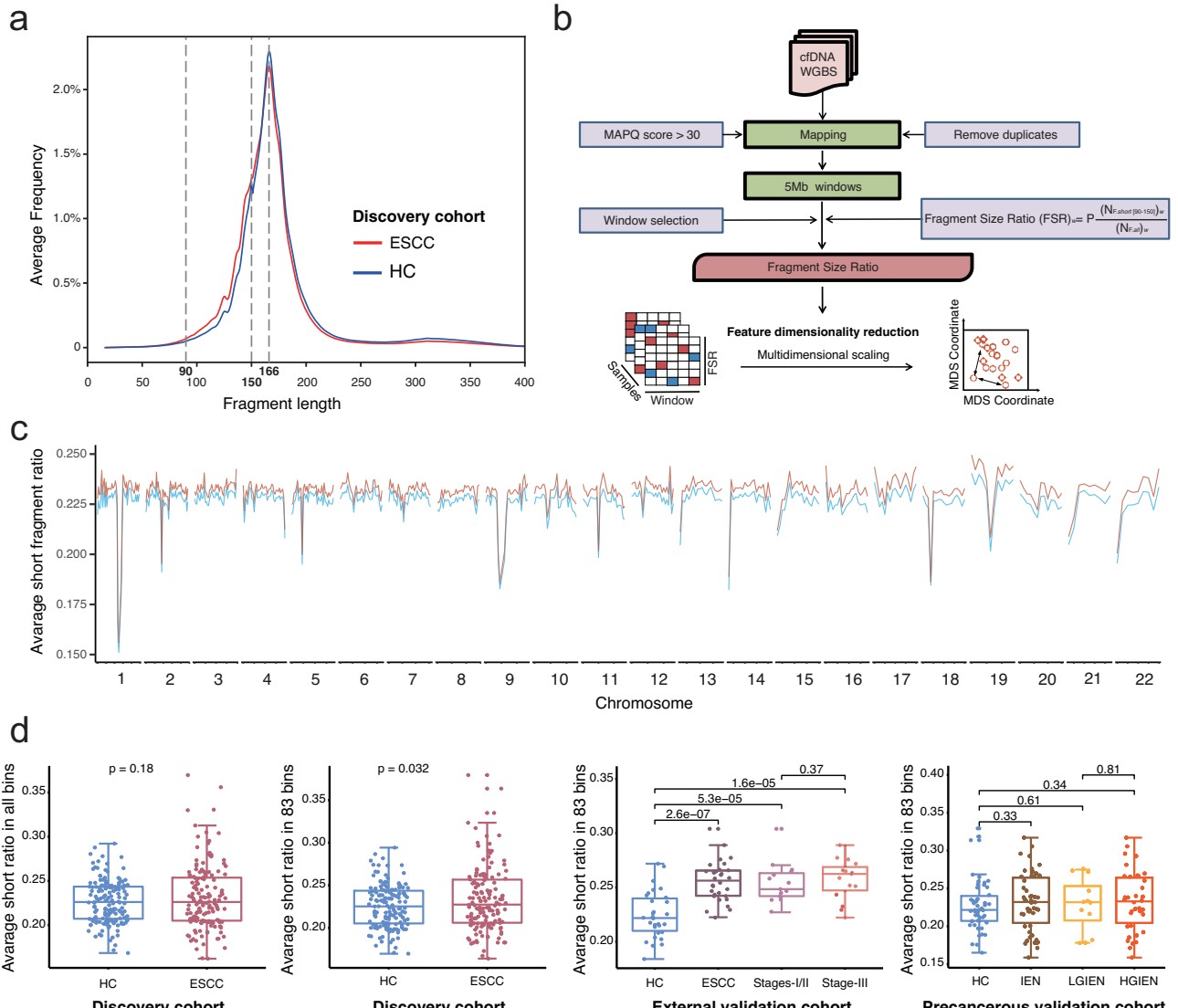

**Fig. 4 | Analyzing the cell-free DNA fragment size in whole-genome bisulfite sequencing data. a** The fragment sizes of cfDNA were surveyed in the discovery cohort. Both the peaks in cfDNA were 166 bp in esophageal squamous cell carcinoma (ESCC) patients and healthy controls. However, more short fragments (90–150 bp) were found in the ESCC groups. **b** The ratio of short fragments in cfDNA was calculated as the fragment size ratio (FSR) in whole-genome bisulfite sequencing data. **c** The human genome was divided into 5 Mb bins, resulting in a total of 1082 (541 bins × 2) FSR features. **d** In the discovery cohort, no significant difference in the average FSR was observed across all bins between ESCC patients and HCs. However, we identified 83 bins where the FSRs were significantly elevated in ESCC patients than HCs in the discovery cohort. The average FSRs in the 83 selected bins were significantly higher in the ESCC patients in the discovery cohort and the external validation cohort, but not the intraepithelial neoplasia patients in the precancerous validation cohort (two-sided Mann–Whitney U-test, $p < 0.05$). Data are presented as median values with maximums and minimums. ESCC esophageal squamous cell carcinoma, IEN intraepithelial neoplasia, LGIEN low-grade IEN, HGIEN high-grade IEN, HC healthy control, WGS whole-genome sequencing, WGBS whole-genome bisulfite sequencing, MAPQ mapping quality, MDS multidimensional scaling. Source data are provided as a Source Data file.

CI: 0.97–1.00 for the DMR *plus* CNV model and AUC = 0.99, 95% CI: 0.98–1.00 for the EMMA model, as opposed to the ESCC-cfMeth score with an AUC = 0.90, 95% CI: 0.87–0.94; $p = 2.5 \times 10^{-7}$ and $5.6 \times 10^{-8}$; Supplementary Fig. 5). Similar superior performance of the DMR *plus* CNV model and the EMMA model was observed in the external validation cohort (AUCs = 0.94 and 0.95, respectively, vs. 0.89 for the ESCC-cfMeth score) and the precancerous validation cohort (AUCs = 0.89 for both combined models and 0.87 for the ESCC-cfMeth score; Fig. 5c).

To achieve a specificity of more than 95%, a cutoff point was selected for the EMMA model. In comparison to the ESCC-cfMeth score, the EMMA model improved sensitivities from 70% to 87% in the external validation cohort and from 50% to 62% in the precancerous validation cohort (Fig. 5c; Supplementary Table 3). Overall, the EMMA

model increased the detection rate to 62% for IENs, 78% for stage I, 83% for stage II, and 93% for stage III ESCCs, while maintaining specificities >95% in two independent validation cohorts (Fig. 5d). In the external validation cohort, the detection rates increased across all stages. FSR contributed to one additional stage-I ESCC, and CNV events were detected in two additional stage-II/III ESCCs. Additionally, the EMMA model identified one stage-I ESCC patient and one stage-III ESCC patient who were not detected by any of the three single-modal models (Fig. 5e). Convincingly, a similar increase in detection rates was also observed in IENs (Supplementary Fig. 5).

To estimate the potential benefits of increased detection rates for early-stage ESCC and precancerous lesions, we conducted an adapted interception analysis using published baseline data of Chinese ESCC patients as an example[6,32–34]. When the ESCC-cfMeth and EMMA

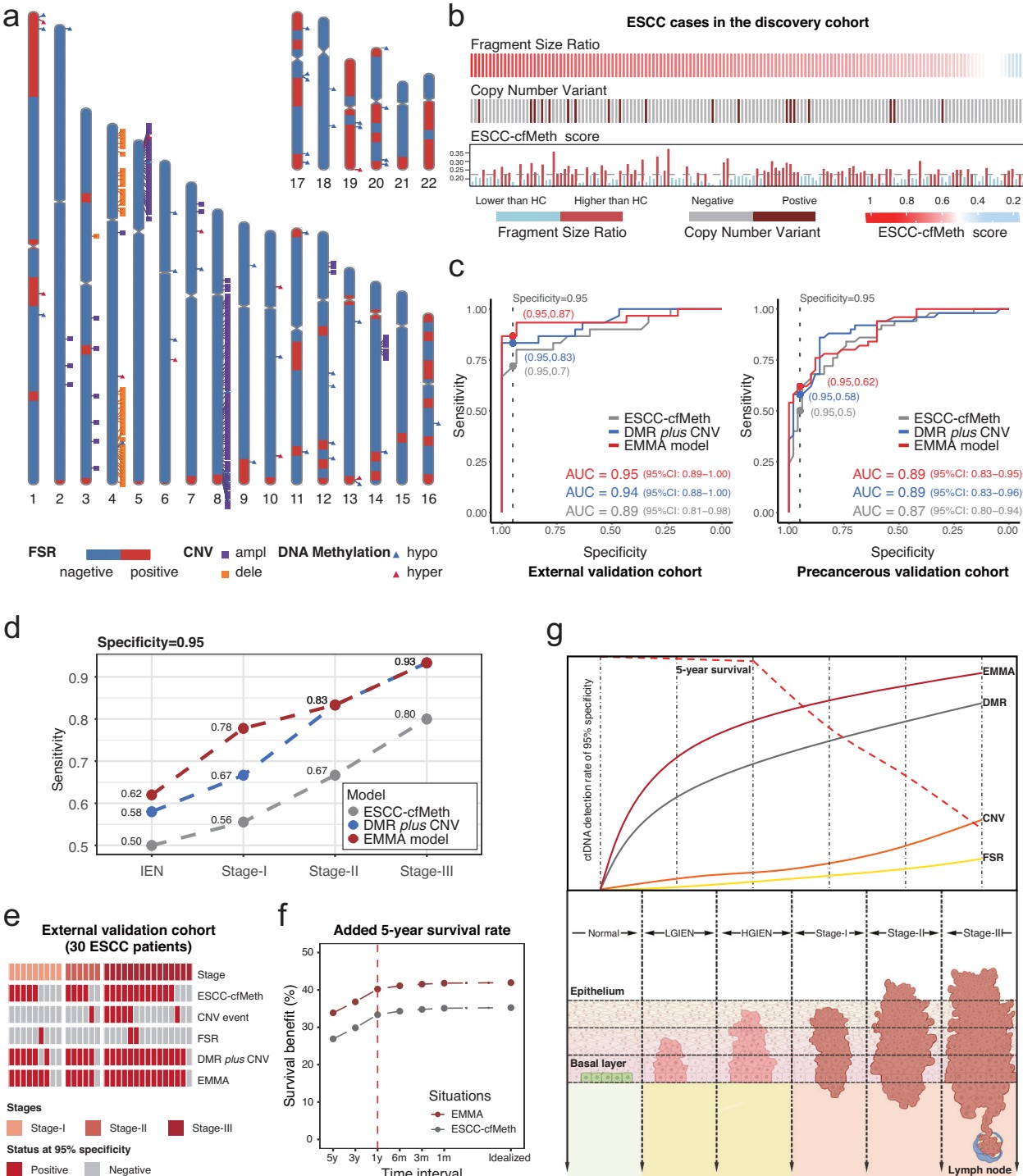

**Fig. 5 | Complementarities of the three cell-free DNA features and the performance of the combined models. a** The distributions of cfDNA methylation markers, copy number variants, and fragmentation features in the human genome. **b** Complementarities between these three features were found in the esophageal squamous cell carcinoma (ESCC) patients in the discovery cohort. Each bar indicates the status of each feature in a ESCC patient. **c** The diagnostic performances of the ESCC-cfMeth score, DMR *plus* CNV model, and EMMA model were evaluated in the external validation cohort and the precancerous validation cohort. **d** The detection rate of the ESCC-cfMeth score, the DMR plus CNV model, and the EMMA model for intraepithelial neoplasia (IEN), stages I, II, and III ESCC. **e** In the external validation cohort, improved performances of the combined models resulted from

the complementarities in three features. **f** The potential survival benefit of the EMMA model and the ESCC-cfMeth model were estimated according to different test intervals, ranging from 5 years to continuous testing (idealized). **g** The Schematic diagram of the detection rates for different cfDNA features and combined models and 5-year survival in different stages. ESCC esophageal squamous cell carcinoma, IEN intraepithelial neoplasia, FSR fragment size ratio, CNV copy number variant, DMR differentially methylated region, EMMA expanded multimodal analysis, Meth methylation, hypo hypomethylation, hyper hypermethylation, AUC area under curve, CI confidence interval. Source data are provided as a Source Data file.

approaches are implemented, the 5-year overall survival rates of ESCC patients in China show potential improvement by early detection and intervention at the IEN stage (Fig. 5f). Comparatively, the EMMA model, which integrates multi-modal data, demonstrates higher survival benefits in the assumed clinical scenario compared to the ESCC-cfMeth model, which solely utilizes cfDNA methylation. Depending on the test interval, ranging from 5 years to continuous testing (idealized), the ESCC-cfMeth model exhibited a potential increase in 5-year overall survival rates by 26.90% to 35.25%. In contrast, the EMMA model shows a potential increase of 33.87% to 41.95%. Specifically, when the EMMA approach is applied annually at the IEN stage, the 5-year overall survival of ESCC patients in China could further increase by 6.9% compared to the ESCC-cfMeth model (40.22% vs. 33.37%). Our findings indicate that cfDNA methylation markers outperform CNV and fragmentation markers, particularly in early stages and precancerous lesions. Multimodal analysis could further enhance their detection performance in a cost-effective and sample-saving manner, potentially improving survival rates (Fig. 5g).

## Biological significance of cfDNA methylation markers

We examined the methylation levels of the 50 DMRs from the ESCC-cfMeth model across 81 common cell types[35], confirming their ESCC-specificity (Supplementary Fig. 6). To evaluate the value of these DMRs in ESCC molecular subtyping, we classified the 155 ESCC patients into three groups based on the average methylation levels of these 50 DMRs using hierarchical clustering in the ECGEA cohort. These groups were labeled methylation-dominant ($n = 69$), methylation-moderate ($n = 54$), and methylation-poor ($n = 32$) (Fig. 6a). No significant differences were found in gender, age, stage, tumor location, or grade among these groups.

In our previous multi-omics analysis of the ECGEA cohort[18], ESCCs were categorized into four subtypes: cell cycle pathway activation (CCA), NRF2 oncogenic activation (NRFA), immune suppression (IS), and immune modulation (IM). The CCA subtype was characterized by recurrent CNVs and the CpG island methylator phenotype (CIMP), while IM cases tended to respond more effectively to immunotherapy than the other subtypes. Notably, we found the methylation-dominant group contained a higher proportion of ESCCs belonging to the CCA subtype (37.68% [26/69]) compared to the methylation-moderate (18.52% [10/54], $p = 0.03$) and methylation-poor (9.38% [3/32], $p = 4.0 \times 10^{-3}$) groups. Conversely, the methylation-dominant group had a lower proportion of ESCCs belonging to the IM subtype (15.94% [11/69]) compared to the methylation-moderate (42.59% [23/54]; $p = 1.2 \times 10^{-3}$) and methylation-poor (43.75% [14/32]; $p = 5.5 \times 10^{-3}$) groups (Fig. 6b). Next, we investigated the TME components in these three groups and found the methylation-dominant group had more epithelial cells and fewer immune cells than the other two groups (all $p < 0.05$; Fig. 6c). These findings highlight the diagnostic methylation markers could be potential biomarkers for distinct molecular subtypes and TME in ESCC.

Subsequently, we examined genes and pathways associated with methylation features by conducting differential expression and gene ontology (GO) analysis of gene expression in ESCC tissues, comparing the methylation-dominant group to the methylation-poor group. Our analysis revealed that 153 genes showed significant differential expression (Benjamini-Hochberg-adjusted $p < 0.05$). The methylation-dominant group showed GO enrichment in cell division-related pathways, including organelle fission, nuclear division, mitotic nuclear division, and chromosome segregation. In contrast, the methylation-poor group exhibited enrichment in immune-related pathways, including T-cell activation, mononuclear cell differentiation, and lymphocyte differentiation (Fig. 6d). Furthermore, the methylation-dominant group displayed better survival outcomes than the other two groups, especially in patients with grade-II ESCC ($p = 0.02$; Supplementary Fig. 7). Thus, the differential gene expression and survival

disparities align with the molecular subtypes and cell components in ESCC.

Previously, esophageal CIMP (E-CIMP) was defined based on 208 promoter sites displaying hypermethylation in over 50% of ESCC samples in the ECGEA cohort[18]. Remarkably, we found the methylation-poor group (12.50% [4/32]) exhibited a low proportion of ESCCs with positive E-CIMP compared to the methylation-dominant (52.17% [36/69]; $p = 1.5 \times 10^{-4}$) and methylation-moderate (38.89% [21/54]; $p = 0.01$; Supplementary Fig. 8). We also surveyed the prevalence of CNVs in 153 selected regions in tissue samples from the ECGEA cohort. Intriguingly, the methylation-dominant group had the highest rate of CNV-positive ESCCs (68.12%, 47/69), likely due to the highly active cell cycle and cell division pathways. However, the methylation-moderate (51.85%; 28/54) and methylation-poor (56.25%; 18/32) groups still exhibited considerable rates of CNV-positive ESCCs, which was not significantly different from the methylation-dominant group (both $p > 0.05$; Supplementary Fig. 8). This observation is consistent with the contribution of CNV features to the EMMA model. Collectively, our findings indicate the potential utility of the cfDNA methylation markers for molecular subtyping and guiding treatment decisions in ESCC.

Mutations in TP53[36,37] and APOBEC genes[38] as well as the APOBEC mutational signatures (SBS2/13)[39] were frequently observed in ESCC patients. However, there were no significant differences in the somatic mutation rates in TP53 and APOBEC genes and the proportion of the APOBEC mutational signatures between the three groups (all $p > 0.05$; Supplementary Fig. 8). The proportion of the APOBEC mutational signatures was also not associated with the status of carrying CNV in ESCC patients. We also investigated the overlapping of TP53 and APOBEC genes with the multimodal markers and found these genes were not within the regions of either DMRs or CNVs (Supplementary Table 4). However, TP53 and APOBEC3 genes were within the regions of FSRs, which indicates a probably higher proportion of ctDNA in cfDNA in these regions. Thus, identifying the mutations in genes like TP53 in cfDNA might further improve the detection rate of ESCC.

## Discussion

Non-mutational epigenetic reprogramming recently emerged as a pivotal hallmark of cancer[40]. Among various epigenetic modifications, DNA methylation is the most extensively studied in humans, serving as a stable biomarker for cancer development and cell-of-origin[35,41,42]. Aberrant DNA methylation specific to disease progression has also been identified as a key characteristic in early-stage cancer[42,43] and even in pre-invasive cancer lesions[20,44]. Extensive profiling studies have revealed complex and distinctive DNA methylation profiles in cancer[45]. Consequently, these profiles are now being incorporated into cancer diagnostic criteria and have the potential to become standard procedures conducted at major medical centers[46–49].

Despite the increased adoption of the cfDNA methylation markers in liquid biopsy as a promising tumor-naïve strategy for cancer early detection, studies on their biological significance and combination with other cfDNA features are still lacking. Our study introduces the EMMA framework and demonstrates its potential for enhancing whole-genome methylation-based cfDNA analysis, which is considered the most sensitive method for cancer detection[16], through a multimodal analysis. We conducted a comprehensive analysis of cfDNA methylation, CNVs, and fragmentation features in cfDNA from individuals with ESCC/IEN and HCs. Our findings reveal a distinct cfDNA methylation pattern that remains highly sensitive throughout all stages of ESCC, including early stages and precancerous lesions (IENs). In contrast, the CNVs and FSRs are related to late-stage ESCC and showed high specificity (e.g., 98%-100% for CNV events). Understanding the complementarity and timing of these cfDNA features will guide the development of cfDNA-based detection strategies tailored to individuals at varying risks across different cancer stages (Fig. 5g). Furthermore, we correlated the cfDNA methylation markers with the

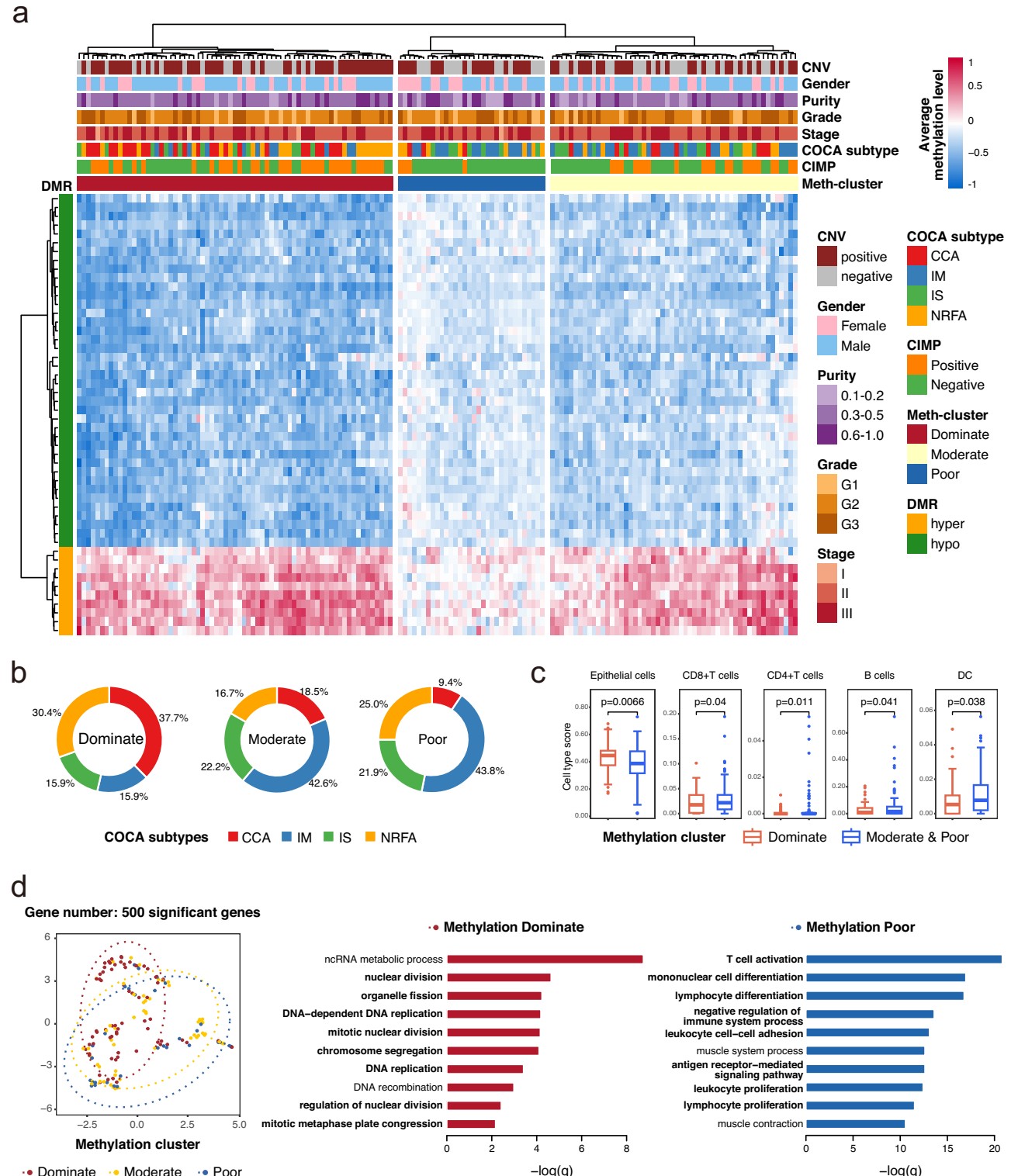

**Fig. 6 | The biological significance of the DNA methylation markers in esophageal squamous cell carcinoma cell-free DNA. a** We divided the 155 patients with esophageal squamous cell carcinoma (ESCC) in the ECGEA cohort into three groups according to the average methylation level of 50 optimal DNA methylation markers in the ESCC-cfMeth score. **b.** The proportions of molecular subtypes of ESCC in the methylation-dominate ($n = 69$), methylation-moderate ($n = 54$), and methylation-poor groups ($n = 32$). **c** The cell components in the tumor micro-environment were compared between the methylation-dominate group ($n = 69$) and the methylation-moderate/poor groups ($n = 86$) by a two-sided $t$ test

($p = 6.6 \times 10^{-3}$, 0.04, 0.01, 0.04, and 0.04 for epithelial cells, CD4 + T cells, CD8 + T cells, B cells, and dendritic cells, respectively). Data are presented as median values with maximums and minimums. **d** The pathways were enriched in the methylation-dominate group and the methylation-poor group. CNV copy number variant, COCA cluster of cluster assignments, CIMP CpG island methylator phenotype, Meth-cluster methylation cluster, DMR differentially methylated region, hypo hypomethylation, hyper hypermethylation, CCA cell cycle pathway activation, IM immune modulation, IS immune suppression, NRFA NRF2 oncogenic activation, DC dendritic cell. Source data are provided as a Source Data file.

molecular subtypes and TME components, expanding their potential application in dynamic monitoring of molecular characteristics and therapeutic decision-making.

For identifying the bona fide ctDNA in cfDNA, multimodal analysis has improved accuracy while remaining cost-effective and sample-saving. However, in the previous CCGA study[16], genetic and fragmentomics features obtained through additional paired targeted sequencing and WGS did not demonstrate complementarity when combined with cfDNA methylation to improve detection sensitivity. In our study, we expanded the genome-wide multimodal analysis of cfDNA to encompass three omics, and our results confirmed the improved performance of the integrated EMMA model. Importantly, EMMA could detect additional cases of early-stage ESCCs and IENs that yielded negative results when using three separate models (Fig. 5e and Supplementary Fig. 4). This improvement may result from increased complementarity of the multimodal information obtained from a single assay. Other studies have also reported elevated detection rates in similar approaches, including combining genome-wide mutational and fragmentation analyses of cfDNA by the GEMINI and DELFI[13] and simultaneous detection of methylation markers and mutations in cfDNA by the Mutation Capsule Plus technology[50].

In a previous study, CNVs were analyzed using WGBS data of 84 ESCC and paired paraneoplastic tissues[51]. Regions with CNVs exhibited higher intratumor DNA methylation heterogeneity than non-CNV regions, indicating a close link between DNA methylation and CNVs. However, limited studies have been conducted on detecting CNVs using WGBS data, potentially due to the alterations and sequence preferences induced by bisulfite treatment. Consequently, identifying CNVs in WGBS data within cfDNA poses greater challenges. In our study, we developed an ESCC-specific approach to identify recurrent CNVs in cfDNA WGBS data, which were consistently observed in both WGS and WGBS data from tissue samples (Fig. 3c). This approach reduced false-positive rates while maintaining high specificity. Thus, the rates of CNV events were notably higher in patients with ESCC and IENs, and they correlated with tumor stages and grades (Fig. 3d).

Although integrating the three cfDNA features enhanced the performance of the EMMA model, they collectively exhibited strong consistency in revealing malignancy. In cfDNA, most optimal DMRs were located in regions with significantly high ratios of short fragment sizes, potentially indicating a higher proportion of ctDNA in these regions (Supplementary Fig. 4). Further, tissue samples from the methylation-dominant group in the ECGEA cohort had the highest prevalence of CNV events, aligning with a previous study on ESCC[52]. This observation is consistent with the enrichment of the CCA subtype in the methylation-dominant group, which is characterized by recurrent CNVs[18].

We investigated the biological significance of identified optimal methylation markers in cfDNA. To begin, DMRs were identified in WGBS data extracted from a large number of paired ESCC tissue samples, and specificity was validated through comparisons with multiple cell types (Supplementary Fig. 5). Within these DMRs, we uncovered known functional genes associated with ESCC displaying significantly different expressions. Subsequently, we categorized ESCC patients into three groups based on the methylation levels of the optimal DMRs. The methylation-dominant group exhibited a higher prevalence of the CCA subtype and epithelial cell components, which also correlated with better survival. In contrast, the analysis of the TME and differential expression showed that the elevated immune cell presence and immune processes may diminish the signal from ESCC-related methylation features (Fig. 6c, d). The methylation-moderate and methylation-poor groups had a higher proportion of ESCC patients belonging to the IM subtype (Fig. 6b). This subtype is more sensitive to immunotherapy than other ESCC subtypes, which indicates the potential of cfDNA testing in predicting and monitoring

immunotherapy responses[53]. Collectively, our study provides a comprehensive insight into the biological and clinical significance of cfDNA methylation markers and offers a non-invasive tool for dynamic monitoring of molecular characteristics in ESCC.

Our study has several limitations. First, despite the analysis of over 1000 whole-genome datasets, our validation cohorts were relatively small. Second, the potential function of these multimodal markers is still unknown, especially in precancerous lesions. Third, we derived CNVs and fragmentation features from WGBS data in a cost-effective and sample-saving manner. However, some cfDNA fragments may be damaged during the bisulfite conversion in WGBS, potentially weakening the signals of CNVs and fragmentation features when compared to those generated directly from WGS data of cfDNA. Fourth, as the mutation calling is unreliable in WGBS data, mutations were not enrolled in this multimodal model. While accurately identifying the mutations in genes like TP53 in cfDNA using additional approaches might further improve the detection rate of ESCC. Furthermore, the survival benefits and prognostic significance of the ESCC-cfDNA and EMMA model require further validation in large-scale real-world cohorts.

In conclusion, we have conducted a comprehensive analysis of cfDNA methylation, CNVs, and fragmentation markers in liquid biopsy to enable ultra-early detection of ESCC. By analyzing cfDNA WGBS data through a multimodal approach, we have identified several staging-specific markers and their complementarity. Our study not only significantly enhanced the non-invasive detection capabilities of ESCC but also holds the potential for dynamic molecular monitoring and treatment guiding.

## Methods
### Ethics approval and consent to participate
This study was reviewed and approved by the relevant ethics committees, including the Institutional Review Boards of Shanxi Medical University and Shanxi Cancer Hospital (ECGEA cohort), the institutional review board and the independent ethics committee of the National Cancer Center, Cancer Hospital, Chinese Academy of Medical Sciences (CHCAMS, the discovery cohort and the precancerous validation cohort), and the ethics committee of the Shanghai Chest Hospital (the external validation cohort). Written informed consent was obtained from each participant.

### Study populations and clinical evaluation
The ECGEA cohort[18] consisted of 155 untreated patients diagnosed with ESCC between May 2017 and July 2018 at Shanxi Cancer Hospital, China. Samples of ESCC tissue and paired adjacent normal/non-neoplastic tissue were collected before any treatment. The discovery cohort comprised 150 patients diagnosed with ESCC or HGIEN of the esophagus and 150 gender- and age-matched HCs from the CHCAMS between May 2019 and December 2022. All the ESCC patients underwent surgery or endoscopic resection. The external validation cohort enrolled 30 patients with ESCC and 30 gender- and age-matched HCs from the Shanghai Chest Hospital between October 2022 and December 2022. The precancerous validation cohort consisted of 50 patients with esophagus IEN who underwent endoscopic resection and 50 gender- and age-matched HCs from the CHCAMS between August 2022 and February 2023.

Each patient's final diagnosis of ESCC or IEN was pathologically confirmed. At least three pathologists reviewed the tumors independently to ensure histological consistency with ESCC and classified them based on WHO criteria. The clinical staging was determined according to the eighth edition of the American Joint Commission of Cancer (AJCC) classification for ESCC[54]. HCs were excluded from any malignant disease through a 6-month follow-up. This study followed the STARD (Standards for Reporting of Diagnostic Accuracy Studies) guidelines[55].

## Whole-genome multi-omic analysis in ESCC tissues and paired adjacent normal tissue

Genomic DNA and total RNA were extracted from the primary tumors and matched adjacent non-neoplastic tissues of 155 ESCC cases from our previous ECGEA cohort[18]. To construct the WGS library, approximately 300 ng of high-quality DNA was fragmented to an average size of 150–200 bp using the Covaris LE220 Sonicator (Covaris, USA), then which was prepared utilizing the TruSeq Nano DNA LT Library Prep Kit (Illumina, USA). For the WGBS library preparation, 200 ng of high-quality DNA was mixed with 1% unmethylated Lambda DNA and fragmented to a 300 bp insert size using the Covaris LE220 Sonicator (Covaris, USA). This was followed by end repair and adenylation. The fragmented DNA was then ligated with methylated adapters. Bisulfite treatment was conducted following the instruction manual of the EZ DNA Methylation-Gold Kit manual (Zymo Research, USA). The resulting single-strand DNA fragments were amplified by polymerase chain reaction (PCR) using the KAPA HiFi HotStart Uracil+ ReadyMix kit (Roche Diagnostics Roche Diagnostics) and purified. For the construction of the RNA library, approximately 1 μg of high/medium-quality RNA was utilized as the input material. Both cytoplasmic and mitochondrial rRNA were removed using the Ribo-Zero Gold kit (Illumina, USA). The rRNA-depleted sequencing libraries from the total RNA were prepared using the Illumina TruSeq Stranded Total RNA Gold preparation kit (Illumina, USA).

The WGS, WGBS, and RNAseq libraries were quantified using the Qubit dsDNA HS Assay (Thermo Fisher Scientific, USA). The size distribution of the resulting sequencing libraries was analyzed using the Agilent BioAnalyzer 2100 (Agilent, USA). Paired-end sequencing was performed using an Illumina NovaSeq6000 following Illumina-provided protocols.

## Blood collection and plasma isolation

Peripheral blood samples (10 mL per person) were collected from all participants before treatment. Blood samples were collected in the EDTA blood collection tubes (Becton Dickinson, USA) and stored at a temperature of 4 °C for no longer than 2 hours before undergoing centrifugations ($1800 \times g$ for 10 min at 4 °C for plasma collection and $16,000 \times g$ for 10 min at 4 °C for removal of cellular debris).

## Cell-free DNA extraction and quantification

The cfDNA was extracted from a median of 2 mL of plasma using the QIAmp Circulating Nucleic Acid Kit (Qiagen, USA) with the Qiagen QIAvac 24 Plus vacuum manifold and QIAvac Connecting System (Qiagen, USA) according to the manufacturer's recommendations. Then, cfDNA was quantified by Qubit 3.0 using the dsDNA HS Assay Kit (Life Technologies, USA). At last, DNA was stored at −80 °C for further analysis.

## Cell-free DNA methylation library preparation

Cell-free DNA (5-50 ng) was spiked with 0.5% unmethylated lambda DNA (Promega, USA), which was sheared by the Covaris S220 instrument (Covaris, USA) to ~350 bp and subjected to bisulfite conversion with EZ DNA Methylation-Lightning Kit (Zymo Research, USA). The converted DNA was processed into library preparation with Accel-NGS Methyl-Seq DNA Library Kit and Methyl-Seq Dual Indexing Kit (Swift Biosciences, USA) according to the manufacturer's protocol.

## Library quantification and whole genome bisulfite sequencing (WGBS) for cell-free DNA

The Prepared libraries were quantified with Qubit dsDNA HS Assay Kit (Life Technologies, USA) and KAPA Library Quantification Kit (KAPA Biosystems, USA), and the library quality was assessed using Agilent 2100 Bioanalyzer (Agilent, USA). Paired-end 150 bp sequencing was performed for each library on the Illumina HiSeq platform to a mean coverage depth of 10× for cfDNA.

## Quality control, data processing, and analysis

Quality control analyses were generated for the raw data using the FastQC (version 0.11.8, www.bioinformatics.babraham.ac.uk/projects/fastqc/). For tissue samples, trim_galore (version 0.6.0, www.bioinformatics.babraham.ac.uk/projects/trim_galore/) was employed to filter low-quality read data and trim adapters, retaining high-quality data with a quality score greater than 20. For cfDNA samples, the bbduk tool from bbmap was utilized to customize a WGBS adapter library and perform adapter trimming to preserve more fragment information. Subsequently, reads were aligned to the hg38 genome using Bismark[56] (version 22.1) to identify the optimal alignment strategy. Duplicate reads were removed using the deduplicate_bismark tool from bismark. Finally, the bismark_methylation_extractor tool was employed to calculate genome-wide methylation levels. The Samtools[57] suite (version 1.9) manipulated alignments in the BAM format. We used bedtools[58] utilities (version 2.30.0) for the comparison, manipulation, and annotation of genomic features in Browser Extensible Data (BED).

## Algorithms for identification of cfDNA methylation markers

In the ECGEA cohort, we extracted DMR and CNV features from WGBS and WGS data in 155 pairs of ESCC tissue samples. These features and the FSRs served as the basis for developing a computational framework for identifying early-stage ESCC patients using plasma cfDNA.

## DMRs feature selection

From 460 cfDNA samples, 300 samples (ESCC: HCs = 150: 150) from the CHCAMS were selected as the training set, which was employed for feature selection and model construction. To mitigate overfitting, all models in this study underwent 10-fold cross-validation using random forest modeling. Initially, a preliminary set of 650 DMR features was selected through an adjusted $p$ value < 0.05 (Wilcoxon test), favoring DMRs with ESCC average values more significant than normal, as determined by the malignant ratio[14]. Subsequently, to minimize costs and reduce the required blood volume for single omics analysis, selecting the minimum number of features that maximally incorporated information about ESCC was essential. Based on the inflection point of model accuracy, the optimal number of features was determined to be 50 (Supplementary Fig. 1). Following, the optimal 50 features (Supplementary Figs. 2 and 3a) were selected using the recursive feature elimination (RFE) strategy for classifier development.

We performed DMR annotation using ChipSeeker[59] and the TxDb.Hsapiens.UCSC.hg38.knownGene database, which includes human gene transcripts and is accessible via Bioconductor. The promoter region was defined as −2 to +2 kb of the transcription start site (TSS).

## Generation of the ESCC-cfMeth scores by machine learning

We constructed a random forest model, the ESCC-cfMeth score, using the 'caret' (version 6.0-93) package with 50 DMRs in the training dataset. The model employed default settings of 500 decision trees, with the number of randomized variables chosen at each split. The model's performance was rigorously assessed using 'ROC' as the evaluation metric. It underwent multiple rounds of meticulous 10-fold cross-validation, iteratively tuning parameters to maximize ROC values, ensuring robust performance evaluation. Ultimately, the model parameters 'mtry' were set as 2. Next, the ESCC-cfMeth score was evaluated in the internal cross-validation, external validation, and precancerous validation sets. To calculate the detection accuracy, samples with ESCC-cfMeth scores ≥ 0.5 were classified as ESCC samples, whereas those below 0.5 were considered non-ESCC/healthy.

## Identification of CNVs in WGBS data

Since no reliable tools are available for calculating CNVs in WGBS data, we developed an algorithm for identifying CNVs in WGBS data inspired

by previous literature on WGS or WGBS data analysis[60,61]. Fragment counting involved standardizing sample data to a uniform depth and segmenting the genome into 1000-kb regions, yielding 2701 regions, with fragment counts quantified within each region. Subsequently, data denoising was performed to address the biased distribution of fragment counts in regions with GC content ranging from 0.3% to 0.6%, which can lead to false positives. A well-established method, locally weighted regression (LOESS), was employed for GC bias correction. The LOESS method employs the following cubic function as weights:

$$B(x) = \begin{cases} (1-x^2)^2, & |x|<1, \\ 0, & |x| \geq 1. \end{cases} \quad (1)$$

$$W(x) = \begin{cases} (1-x^3)^3, & |x|<1, \\ 0, & |x| \geq 1. \end{cases} \quad (2)$$

The x is the CG rate. The selection of weight functions involves using the W function (cubic function) in the first iteration and subsequently employing the B function (quadratic function)

Employ weighted regression to derive the model:

$$\hat{y} = X(X^T W X)^{-1} X^T W y \quad (3)$$

The y is the raw fragment number, and $\hat{y}$ is the predictive fragment number.

In the final step, CNV regions were identified using the GC-corrected fragment counts, calculated according to a specific formula. The model classification was executed using Gaussian mixture models from the 'mclust' package (version 5.4.5, https://github.com/Japrin/mclust) to cluster the log2Ratio values, and hidden Markov models (HMM) from the 'HMM' package (version 1.0.1, https://CRAN.R-project.org/package=HMM) were employed to refine the clustering results.

$$\log_2 Ratio = \log_2 \frac{F_t}{F_n} \quad (4)$$

Where $F_t$ is the fragment number in the tumor regions, $F_n$ is the fragment number in the normal regions.

For cfDNA CNV analysis, due to the diverse ctDNA components in cfDNA samples, a previously established method was employed to extract ctDNA counts within 1000 kb regions, serving as input for ichorCNA[62]. To retain ESCC tissue information, only recurrent CNV regions (present in three or more tissue samples) were analyzed in cfDNA. In the recurrent regions, CNV events were compared between the cfDNA in ESCC patients and those in the HCs from the discovery cohort using a simple Wilcoxon test ($p < 0.05$). Finally, 153 regions were selected with significant differences.

## Fragment size profiling
The FSR profile examines the ratio of short fragments within the human genome. As defined in previous reports, short fragments fall within 100-150 bp[27,46,47]. The human autosomes were partitioned into non-overlapping 5 Mb bins, and within these bins, the ratios of short fragments were computed, resulting in a total of 570 FSR features. A student's t test was employed with a significance threshold of $p < 0.05$ to identify differential features for input into the machine learning algorithms, resulting in 83 FSRs.

However, incorporating these numerous but information-deficient features directly into the model, even with lower weights, could lead to decreased model performance. Therefore, feature dimensionality reduction was performed using Multidimensional Scaling (MDS), a data dimensionality reduction and visualization method that transforms high-dimensional data into a lower-dimensional space (e.g., two or three dimensions). MDS preserves

the distance relationships between data points while allowing for intuitive data observation and analysis. Subsequently, 83 FSRs were reduced to 2 dimensions (stress=1.38). Afterward, a 10-fold cross-validation random forest model was established in the training set using the two-dimensional FSRs, followed by validation on two independent test sets.

## Generation of the DMR plus CNV model and the EMMA model
Furthermore, we incorporated the CNVs and FSRs information to enhance the performance of the ESCC-cfMeth model. Similar to the feature dimensionality reduction process in FSRs, CNV features in 153 regions were also reduced to two dimensions using the MDS. Finally, separate random forest models were established with 10-fold cross-validation for CNVs, FSRs, the DMRs *plus* CNVs model, and the EMMA model (namely, the DMRs *plus* CNVs *plus* FSRs model).

## Clinical benefits estimation
We adopted an interception model designed by Hubbell et al. to evaluate the potential clinical benefits of our diagnostic model in the real-world setting[32]. The clinical benefits were estimated according to current stage-specific diagnostic yields of ESCC[33], the average annual rate of progression from IEN to ESCC[6,34], the shift rates of stages in ESCC[32], and the 5-year overall survival rates of IENs[6] and ESCCs of different stages[33]. The code for the original interception model is available at https://github.com/grailbio-publications/Hubbell_CEBP_InterceptionModel.

## Calculating the average DNA methylation levels
An internal toolset called 'meme_tools' was developed by us, which comprises two major utilities: the towig and the bed_mean_methy. meme_tools can convert files in various formats into methylation level (0–1) wig files and calculate the average methylation levels within specified regions. In this study, towig was used to convert compressed bed files containing single-base methylation levels from 155 pairs of tissue samples derived from Bismark into wig files. Subsequently, bed_mean_methy was employed to calculate the average methylation levels within 650 DMRs and 50 selected DMRs. Running these calculations on a tissue sample is highly efficient, with minimal resource requirements (only 10 GB of memory) and a runtime of approximately 3 minutes. The same procedures were applied to calculate average methylation levels within 81 previously published cell-type-specific regions (Supplementary Fig. 5), except for removing missing CG sites before conversion into wig files.

## Cell type estimation
Cell type deconvolution scores, including epithelium cells, immune cells (including CD8 + T cells, CD4 + T cells, and macrophages), and stroma cells (including endothelial cells and fibroblasts), were calculated by R package 'xcell' (version 1.1.0)[63] with the default gene signatures and "RNA-seq" mode.

## Differential expression and gene set enrichment analysis in RNAseq data
Hierarchical clustering was utilized to stratify subjects based on the DNA methylation levels of 50 DMR features within 155 ESCC tissue-based datasets to discern distinct population groups characterized by varying DNA methylation states. Both samples and DMRs were clustered using the Euclidean distance and Ward's agglomeration method. To identify differentially expressed genes among populations with different DNA methylation states, we employed the Wilcoxon Rank-Sum Test to identify genes that exhibited significant upregulation or downregulation in the respective DNA methylation state pairs.

For differential expression analysis between the methylation-dominant and methylation-poor clusters, R package 'DESeq2' (version 1.32.0)[64] was utilized. Afterward, the *lfcShrink* function was used for log

fold change corrections. A total of 3221 significant genes (Padj<0.1, BH method) were selected, in which 920 genes were upregulated in the methylation-dominant cluster while 2301 genes were upregulated in the methylation-poor cluster. These genes were subjected to further enrichment analysis using the R package 'clusterProfiler' (version 4.0.5)[65] with default parameters. For dimensionality reduction, the top 500 variable genes were selected for principal component analysis (PCA), and the first two PCs were reserved. Then, clustering was performed using the Louvain algorithm.

GO enrichment analyses were performed by the enrichGO function in the R package 'clusterProfiler' (version 3.10.1)[66]. Only the term "biological process" was selected for GO enrichment analysis. Enrichment with FDR was adjusted by Benjamini-Hochberg multiple comparisons.

## Statistical analysis

The receiver operating characteristic (ROC) curves were constructed using the 'pROC' package (version 1.18.0)[67]. The sensitivity, specificity, positive predictive value (PPV), negative predictive value (NPV), accuracy, and the corresponding 95% CI were also calculated using the 'pROC' package. Kaplan-Meier analysis and log-rank tests were conducted using the R-package 'survival'[68] for the overall survival analysis. All statistical analyses (including student's $t$ test, Wilcoxon, and ANOVA) were performed in R (version 4.2.0)[69]. Significance was determined with a threshold of $p < 0.05$. Bar charts, bar plots, pie charts, line graphs, and other visualizations were generated using the 'ggplot2' package (version 3.4.2). Heatmaps were created using the 'ComplexHeatmap' package (version 2.14.0).

## Reporting summary

Further information on research design is available in the Nature Portfolio Reporting Summary linked to this article.

## Data availability

The WGBS data from 460 cfDNA samples generated in this study have been deposited in the Genome Sequence Archive (GSA) for Human database under accession number HRA006113. According to the GSA-human guidelines, the data are available for all non-profit use through a request to the data access committee or the corresponding author (Prof. Zhihua Liu, email: liuzh@cicams.ac.cn) with responses addressed within 14 working days. After access has been granted, the data is available for one year. The multi-omics genome-wide data from tissue samples is available through the GSA database for Human database under accession code HRA003107 (WGS & RNA-seq, https://ngdc.cncb.ac.cn/gsa-human/browse/HRA003107) and HRA003533 (WGBS, https://ngdc.cncb.ac.cn/gsa-human/browse/HRA003533). Source data are provided with this paper.

## Code availability

The code was provided in an open-source repository in Github (https://github.com/packageandcode/EMMA)[70].

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

## Acknowledgements

This study was supported by funding from the National Natural Science Foundation of China (82030089 to Z.L., 82188102 to Z.L., 82272938 to J.L.), the National Key R&D Program of China (2021YFC2501000 to Z.L., 2020YFA0803300 to Z.L.), the CAMS Innovation Fund for Medical Sciences (2021-I2M-1-018 to Z.L., 2021-I2M-1-067 to H.C.), the Fundamental Research Funds for the Central Universities (3332021091 to Z.L.), and Beijing Nova Program (20220484059 to J.L.). We thank all the individuals, families, and physicians involved in the study for their participation.

## Author contributions

Z.L. and J.S. conceived and administratively supported this study. J.L., Q.W., H.C., Z.L., Z.L., T.G., W.S., M.L., N.Z. and J.C. collected the samples and the patient data. J.S., L. D., C.L., J.L. and H.X. performed data cleaning and statistical analysis. L.D., C.L., J.L., X.W. and H.X. devised the algorithm and performed data analysis and interpretation. J. L., L.D., Q.W., C.L., H.C., J.S. and Z.L. wrote the manuscript. J.L., L.D., C.L., H.X., Z.J. and T.S. drew the schematic diagrams. All the authors have read and approved the final manuscript for publication.

## Competing interests

The authors declare no competing interests.
