## [Peer Review File · Nature Communications]

Multimodal analysis of cfDNA methylomes for early detecting esophageal squamous cell carcinoma and precancerous lesionsREVIEWER COMMENTS

Reviewer #1 (Remarks to the Author): expert in ESCC genomics

This approach is fundamental and cutting edge- to demonstrate how cancer patients can be interrogated to identify not only the molecular basis for development/heterogeneity but also for diagnostic tools using plasma/blood. The study summarizes a very comprehensive and detailed analysis of multi-dimensional features of cancer-dependent biomarkers present in liquid biopsies. Specifically, the authors evaluate several features of ctDNA from cfDNA samples; including fragmentation features from whole genome bisulfite sequencing, CNVs, and methylation status. The work is well controlled, including over 100 oesophagus cancer patients and 100 'controls' from normal population.

Overall, these types of multi-dimensional studies using machine learning are very difficult to judge and even accept as 'true;' because the ability to 'reproduce' the data (even an individual figure) are highly dependent on tools and expertise that is rather limiting, globally. Nevertheless, although there are few laboratories that can reproduce these data in a timely manner (using blood/plasma from any cancer type), the blueprint presented by the authors demonstrates how key biomarkers identified in this study, such as DMRs, can be used in esophagus cancer patient cohorts. In addition, we would hope that many cancer types, by the blueprint reported in this study, that can be processed to collect a library of such high value as the authors have done with these cancer types. This study does represent part of the future of cancer annotation and is aspirational.

Some minor comments.

1. In Figure 3, amplification events on chromosome 3, 8, and 11 in cancer v normal controls (as examples) is quite striking; do these chromosomes contain any known drivers that can be predicted or known from prior studies, or does the use of cfDNA and the tools defined by this study, allow completely novel discoveries?
2. Is EMMA really superior to Meth only (Fig 5G)? Multimodal analysis though marginally better, may not be practical if other laboratories are to use this blueprint on other cancer types. Is the incorporation of CNV plus methylation basically, additive? Related to this, the information is generally statistical—how can this be used with confidence to define some clinical stage on a personalized level? Can it be used in a patient specific manner to diagnose individuals?
3. As another minor comment, there is some discussion in literature whether methylation is relevant for cancer and the authors could mention this to highlight the significance of these current approaches.
4. Figure legends are not well annotated making the work not easy to read.

Reviewer #2 (Remarks to the Author): expertise in DNA methylation bioinformatics

In the manuscript entitled "Expanded multimodal analysis of cell-free DNA methylomes for ultra-early detection of esophageal squamous cell carcinoma and precancerous lesions' ". Firstly, the authors showed the functional relevance of DNA methylation, Copy number variations (CNV) and short fragments ratio (fragment size ratio, FSR) in cfDNA with ESCC and precancerous lesions. Furthermore, they developed a machine-learning based modal called EMMA, simultaneously considering differential DNA methylation regions(DMR), CNV and FSR information from the whole genome bisulfite sequencing data of cell-free DNA from 230 non-metastatic esophagus squamous cell carcinoma or precancerous lesions patients' plasma samples and 230 matched healthy control plasma samples. The authors claimed that EMMA has both higher sensitivity and specificity than those using only DMR, DMR combined with CNV data. At last, they further classified the 155 patients samples into 3 groups based on cfDNA methylation status hyper, moderate and hypo methylation group. And found that the cfDNA methylation groups correlates with the subtypes, cell type enrichment and gene expression activations of the ESCC tissues, further suggesting the potential of cfDNA methylation as an important factor for ESCC subtyping and treatment guidance. Below are some questions that need to be clarified.

1. It's a large patient sample cohort in the manuscript. Have the patients undergone any medical

treatments before they do blood draw or tissue sampling?

2. Could you make detailed explanations for some figures? For example. In Fig.2a, what is the difference between the left and right panel?
3. In Fig.2e, the authors showed the differential expression of a few classical genes between ESCC and HC samples. How about those in the precancerous cohort?
4. Gene mutations such as TP53 and APOBEC was frequently found in ESCC patients (PMID: 36266286, PMID: 34663923,). What is the gene mutations/ SNV status in cfDNA DMR, CNV and FSR regions? Could you check if the TP53 and APOBEC mutations are in the DMR?
5. Could you add the WGS and RNA-seq library preparation for the tissues in the method part?
6. In Fig. 5c, the AUC of EMMA model didn't outperform that much compared with that using the DMR plus CNV model. Have you tried to change the short fragments size range other than 90 to 150bp for the EMMA model? If combined with TP53 and APOBEC mutation status from the cfDNA, what is the performance of EMMA look like?
7. The data in the manuscript can't fully support the title that the EMMA model enables ultra-early detection of ESCC. For example, from Fig.5d, the EMMA model's performance becomes better with the progress of the tumor stages. Please consider changing "ultra early" to a moderate description of the modal.

Reviewer #3 (Remarks to the Author): expertise in ESCC clinical biomarkers

Understanding the combined performances and biological significance of different cfDNA features is important to exploring non-invasive methods for cancer early detection. In this study, Liu et al. reported a novel computational framework for the early diagnosis of ESCC through multi-modal analysis of whole-genome methylation data from cfDNA. The authors systematically revealed the timing and complementarity of different cfDNA features and obtained high detection rates of precancerous lesions and early-stage disease. Moreover, the diagnostic DNA methylation features were found to be associated with distinct molecular subtypes and tumor microenvironments. This study also provides a high-quality dataset of cfDNA methylome, which would further benefit future studies on liquid biopsy of ESCC. Overall, this is a solid study with a meticulous design, stringent analysis, and novel findings. However, there are still some minor concerns that need to be clarified.

1. To estimate the potential benefits to 5-year overall survival rates of ESCC by the multi-modal cfDNA model, the authors conducted an adapted interception analysis. The detailed baseline data used in the analysis and their sources should be provided in the method.
2. To demonstrate the advantages of the multi-modal approach, the potential survival benefits should be compared between the EMMA model and the cfDNA-cfMeth model in at least one assumed clinical scenario.
3. As the authors stated in the limitation, the external validation cohort is relatively small. How was the sample size for the external validation cohort determined?
4. The cut-off value of 0.5 for the ESCC-cfMeth score was used to differentiate between ESCC and normal samples. How was this value set?
5. Both the abbreviations "EIN" and "IEN" are used in the manuscript and figures, which is unclear.
6. The web links for the data and code generated in this study should be provided in the "Availability of data and materials".

RESPONSE TO REVIEWERS' COMMENTS

Reviewer #1 (Remarks to the Author): expert in ESCC genomics

This approach is fundamental and cutting edge- to demonstrate how cancer patients can be interrogated to identify not only the molecular basis for development/heterogeneity but also for diagnostic tools using plasma/blood. The study summarizes a very comprehensive and detailed analysis of multi-dimensional features of cancer-dependent biomarkers present in liquid biopsies. Specifically, the authors evaluate several features of ctDNA from cfDNA samples; including fragmentation features from whole genome bisulfite sequencing, CNVs, and methylation status. The work is well controlled, including over 100 oesophagus cancer patients and 100 'controls' from normal population.

Overall, these types of multi-dimensional studies using machine learning are very difficult to judge and even accept as 'true;' because the ability to 'reproduce' the data (even an individual figure) are highly dependent on tools and expertise that is rather limiting, globally. Nevertheless, although there are few laboratories that can reproduce these data in a timely manner (using blood/plasma from any cancer type), the blueprint presented by the authors demonstrates how key biomarkers identified in this study, such as DMRs, can be used in esophagus cancer patient cohorts. In addition, we would hope that many cancer types, by the blueprint reported in this study, that can be processed to collect a library of such high value as the authors have done with these cancer types. This study does represent part of the future of cancer annotation and is aspirational.

Response: Thank you very much for your crucial comment and the affirmation on our manuscript: "This approach is fundamental and cutting edge..." and "This study does represent part of the future of cancer annotation and is aspirational".

Please see our point-to-point responses to specific comments for improvement below.

Some minor comments.

1. In Figure 3, amplification events on chromosome 3, 8, and 11 in cancer v normal controls (as examples) is quite striking; do these chromosomes contain any known drivers that can be predicted or known from prior studies, or does the use of cfDNA and the tools defined by this study, allow completely novel discoveries?

Response: Thank you for your constructive comment. There are 583 genes located in the regions of CNV markers in the ESCC cfDNA. Of those genes, 27 and 301 are amplified on chromosomes 3 and 8, respectively, while 120 are deleted on chromosome 4. However, even though our previous whole genome sequencing studies showed that 11q13.3 amplification containing *CCND1* is recurrent in ESCC tissues (Cui Y, *et al. Cell Res.* 2020;30[10]:902-913; PMID: 32398863), no amplification event was observed on chromosome 11 in the ESCC cfDNA. This could be attributed to the damage inflicted on certain cfDNA fragments by the bisulfite conversion process during WGBS, further attenuating the signals of CNVs in these regions. Thus, only 38.19% of amplifications (388/1016) and 17.17% of deletions (198/1153) which were identified in WGS and WGBS data from tissue samples were successfully recalled in cfDNA. We've stated this in the

limitation as follows. “**However, some cfDNA fragments may be damaged during the bisulfite conversion in WGBS, potentially weakening the signals of CNVs and fragmentation features when compared to those generated directly from WGS data of cfDNA.**” (Page 20, Lines 20-22)

To ascertain whether there are known drivers among those genes affected by CNVs, we annotated these genes using oncogenes from the ONGene database (Liu Y, *et al. J Genet Genomics*. 2017;44[2]:119-121; PMID: 28162959), recognized for their potential to induce cancer. Our findings unveiled the amplification of the oncogene *TP63* on chromosome 3, as well as amplifications of 30 other oncogenes on chromosome 8, including *MYC* and *KAT6A*. These results indicate that the amplification events on chromosomes 3 and 8 have a significant impact on some known driver genes, potentially playing a crucial role in the development of ESCC.

To reveal the potential function of these genes, we analyzed the association between the expression levels of these genes with the prognosis in the tissue samples in our previous ECGEA cohort (Liu, Z. *et al. Cancer Cell*. 2023;41:181-195.e189; PMID: 36584672). We listed these genes and their impact on ESCC prognosis in **the newly added Extended Data Table 4** in the revised manuscript.

As a result, 41 genes were found to have significant associations with poor prognosis in ESCC patients ($p < 0.05$ and $HR > 1$), including 19 genes within amplifications in chromosome 8 and 8 genes within deletions in chromosome 4. Some of these genes were reported to be associated with late-stage events in cancer including immune evasion, treatment resistance, and metastasis. For example, *FBXO32* within the amplification in chromosome 8 is reported to be amplified in metastatic cancers, which is essential for tumor growth and metastasis (Sahu SK, *et al. Nat Commun*. 2017;8[1]:1523; PMID: 29142217). *FOXA1*, which is located within the amplification in chromosome 14, its overexpression suppresses the immune response and could be a prognostic factor to predict immuno- and chemotherapy resistance in several types of cancer (He Y, *et al. J Clin Invest*. 2021;131[14]:e147025; PMID: 34101624). Although the functions of most of these genes have not yet been proven in ESCC, the association of these genes' expression with prognosis indicates that they may play a significant role in cancer progression. On the other hand, higher proportions of CNV events were found in the methylation-dominant subtype (Extended Data Figure 6a). Thus, the CNVs are not only the trigger events but might also be an endophenotype in this ESCC subtype driven by the cell cycle pathway activation (CCA) pathway.

We added these results to the “**Identification of CNVs in cfDNA for ESCC detection**” in the **Result** section in the revised manuscript as follows.

“There are 583 genes located in the regions of CNV markers in the ESCC cfDNA. Of those genes, 27 and 301 are amplified on chromosomes 3 and 8, respectively, while 120 are deleted on chromosome 4. However, even though our previous whole genome sequencing studies showed that 11q13.3 amplification containing *CCND1* is recurrent in ESCC tissues²¹, there was no amplification event was observed on chromosome 11 in the ESCC cfDNA. This could be attributed to the damage inflicted on certain cfDNA fragments by the bisulfite conversion process during WGBS, further attenuating the signals of CNVs in these regions. To reveal the potential function of these genes, we annotated these genes using oncogenes from the ONGene database³⁰ and

analyzed the association between the expression levels of these genes with the prognosis in the tissue samples in our previous ECGEA cohort²⁰ (Extended Data Table 4). As a result, 41 genes were found to have significant associations with prognosis in ESCC patients, including nine-teen genes with amplifications in chromosome 8 and eight genes with deletions in chromosome 4. Some of these genes were reported to be associated with late-stage events in cancer including immune evasion, treatment resistance, and metastasis, such as *FBXO32*³¹ within the amplifications in chromosome 8 and *FOXA1*³² within the amplifications in chromosome 14.” (Page 11, Lines 4-18)

References in this section:

20. Liu, Z. et al. Integrated multi-omics profiling yields a clinically relevant molecular classification for esophageal squamous cell carcinoma. *Cancer Cell* 41, 181-195.e189 (2023). <https://doi.org:10.1016/j.ccell.2022.12.004>
21. Cui, Y. et al. Whole-genome sequencing of 508 patients identifies key molecular features associated with poor prognosis in esophageal squamous cell carcinoma. *Cell Res* 30, 902-913 (2020). <https://doi.org:10.1038/s41422-020-0333-6>
30. Liu, Y., Sun, J. & Zhao, M. ONGene: A literature-based database for human oncogenes. *J Genet Genomics* 44, 119-121 (2017). <https://doi.org:10.1016/j.jgg.2016.12.004>
31. Sahu, S. K. et al. *FBXO32* promotes microenvironment underlying epithelial-mesenchymal transition via CtBP1 during tumour metastasis and brain development. *Nat Commun* 8, 1523 (2017). <https://doi.org:10.1038/s41467-017-01366-x>
32. He, Y. et al. *FOXA1* overexpression suppresses interferon signaling and immune response in cancer. *J Clin Invest* 131 (2021). <https://doi.org:10.1172/jci147025>

2. Is EMMA really superior to Meth only (Fig 5G)? Multimodal analysis though marginally better, may not be practical if other laboratories are to use this blueprint on other cancer types. Is the incorporation of CNV plus methylation basically, additive? Related to this, the information is generally statistical—how can this be used with confidence to define some clinical stage on a personalized level? Can it be used in a patient specific manner to diagnose individuals?

Response: Thank you for your professional comment. The EMMA model demonstrated significantly superior performances than the ESCC-cfMeth score in the discovery cohort (AUCs=0.99 vs. 0.90, $p=5.6\times 10^{-8}$ in the 10-fold cross-validation; Extended Data Figure 4) and the external validation cohort (AUCs= 0.95 vs. 0.89, $p=0.045$; **Fig. 5c**). However, the performance of EMMA was superior but without significance than that of the ESCC-cfMeth score in the precancerous validation cohort (AUCs=0.89 vs. 0.87, $p=0.8075$). It might result from that the CNV and FSR features contribute mainly in late-stage ESCC.

To identify the true diagnostic features for ESCC, we developed the EMMA framework based on a “tissue-cfDNA-tissue” strategy. Thus, the whole-genome blueprint of DMRs and CNVs in cancer tissue is necessary for developing this multimodal analysis. Hopefully, with the exponential growth in data volume on cancer-specific genetics and epigenetics, these multimodal approaches might have a chance to be applied to more cancer types and pan-cancer early detection.

The combined effects of CNV and DNA methylation are not simply additive. They were combined in a machine-learning manner. First, to reduce the dimensionality of the CNV features

in 153 regions, we created two composite parameters to stand for the CNV using Multidimensional Scaling (MDS). Next, we developed the combined models through separate random forest models in the discovery cohort with 10-fold cross-validation. We apologize for previously describing these only in the **Method** section. We've added this description in the **Result** section in the revised manuscript as follows.

“To reduce the dimensionality of the CNV features in 153 regions and FSRs in the 83 bins, we created two composite parameters for each modal (**Fig. 1a**). Next, we developed two combined diagnostic models through separate random forest models in the discovery cohort with 10-fold cross-validation: one combining the 50 DMRs with two CNV parameters (DMR plus CNV model) and another combining the 50 DMRs with two CNV parameters and two FSR parameters (EMMA model).” (Page 13, Lines 5-10)

The ESCC-cfMeth score and the combined EMMA models are patients-naïve manners like most cfDNA methylation-based and fragmentomics-based approaches. Based on large-scale ESCC cohort data from multiple centers, a robust model has been established. Through extensive model training, consistent parameters and cut-off values have been determined. As a result, when these models are applied to diagnose other samples or individuals, the outcomes are stable and reliable. Although clinical staging was not considered during the model development, the characteristics of cfDNA are more significant in the later stages of the disease. Therefore, the likelihoods which were predicted by the ESCC-cfDNA score and the EMMA model are increased with the stages of ESCC ($p=1.81\times 10^{-6}$; **Figure R1**).

Figure R1. The scores of the EMMA model were elevated with ESCCs stages.

Abbreviation: EMMA, expanded multimodal analysis; IEN, intraepithelial neoplasia.

3. As another minor comment, there is some discussion in literature whether methylation is relevant for cancer and the authors could mention this to highlight the significance of these current

approaches.

Response: Thank you for your helpful suggestion. According to your suggestion, we've added a discussion on the relevance of DNA methylation and cancer and the significance of the cfDNA methylation-based approaches in clinical practice in the **Discussion** section as follows.

“Non-mutational epigenetic reprogramming recently emerged as a pivotal hallmark of cancer⁴¹. Among various epigenetic modifications, DNA methylation is the most extensively studied in humans, serving as a stable biomarker for cancer development and cell-of-origin^{37,42,43}. Aberrant DNA methylation specific to disease progression has also been identified as a key characteristic in early-stage cancer^{43,44} and even in pre-invasive cancer lesions^{22,45}. Extensive profiling studies have revealed complex and distinctive DNA methylation profiles in cancer⁴⁶. Consequently, these profiles are now being incorporated into cancer diagnostic criteria and have the potential to become standard procedures conducted at major medical centers⁴⁷⁻⁵⁰.” (Page 17, Lines 14-21)

References in this section:

22. Lin, D. C., Wang, M. R. & Koeffler, H. P. Genomic and Epigenomic Aberrations in Esophageal Squamous Cell Carcinoma and Implications for Patients. *Gastroenterology* 154, 374-389 (2018). <https://doi.org:10.1053/j.gastro.2017.06.066>
41. Hanahan, D. Hallmarks of Cancer: New Dimensions. *Cancer Discov* 12, 31-46 (2022). <https://doi.org:10.1158/2159-8290.Cd-21-1059>
42. Davalos, V. & Esteller, M. Cancer epigenetics in clinical practice. *CA Cancer J Clin* 73, 376-424 (2023). <https://doi.org:10.3322/caac.21765>
43. Dor, Y. & Cedar, H. Principles of DNA methylation and their implications for biology and medicine. *Lancet* 392, 777-786 (2018). [https://doi.org:10.1016/s0140-6736\(18\)31268-6](https://doi.org:10.1016/s0140-6736(18)31268-6)
44. Baylin, S. B. & Jones, P. A. A decade of exploring the cancer epigenome - biological and translational implications. *Nat Rev Cancer* 11, 726-734 (2011). <https://doi.org:10.1038/nrc3130>
45. Teixeira, V. H. et al. Deciphering the genomic, epigenomic, and transcriptomic landscapes of pre-invasive lung cancer lesions. *Nat Med* 25, 517-525 (2019). <https://doi.org:10.1038/s41591-018-0323-0>
46. Koch, A. et al. Analysis of DNA methylation in cancer: location revisited. *Nat Rev Clin Oncol* 15, 459-466 (2018). <https://doi.org:10.1038/s41571-018-0004-4>
47. Yousefi, P. D. et al. DNA methylation-based predictors of health: applications and statistical considerations. *Nature Reviews Genetics* 23, 369-383 (2022). <https://doi.org:10.1038/s41576-022-00465-w>
48. Papanicolau-Sengos, A. & Aldape, K. DNA Methylation Profiling: An Emerging Paradigm for Cancer Diagnosis. *Annu Rev Pathol* 17, 295-321 (2022). <https://doi.org:10.1146/annurev-pathol-042220-022304>
49. Schrag, D. et al. Blood-based tests for multicancer early detection (PATHFINDER): a prospective cohort study. *Lancet* 402, 1251-1260 (2023). [https://doi.org:10.1016/s0140-6736\(23\)01700-2](https://doi.org:10.1016/s0140-6736(23)01700-2)
50. Vermeulen, C. et al. Ultra-fast deep-learned CNS tumour classification during surgery. *Nature*

4. Figure legends are not well annotated making the work not easy to read.

Response: Thank you very much for your suggestion. We've rewritten all the figure legends for Figures 1-6 and Extended Data Figures 1-8 as shown below.

Figure 1. Study design and patient enrollment.

a. An approach called 'expanded multimodal analysis' (EMMA) has been developed using machine learning to enhance the detection of ctDNA from cfDNA in plasma samples. This is achieved by comprehensively analyzing cancer-derived differentially methylated regions (DMRs), copy number variants (CNVs), and fragmentation features in the cfDNA whole-genome bisulfite sequencing (cfWGBS) data. The cancer-derived DMRs and CNVs were initially identified from paired WGBS and whole-genome sequencing (WGS) data of primary tumors and matched adjacent non-neoplastic tissues of 155 patients with esophageal squamous cell carcinoma (ESCC). Subsequently, the ESCC-derived DMRs and CNVs were examined in cfWGBS data and further utilized with the proportion of short cfDNA fragment sizes to train the diagnostic models in the discovery cohort. The performance of each diagnostic model was independently assessed in an external ESCC cohort and a precancerous cohort. To unveil the biological significance of these optimal DMRs, we correlated them with multi-omics-based molecular subtypes and transcriptomic profiles in the paired ESCC tissue samples. **b.** The discovery cohort encompassed 150 patients with ESCC or high-grade intraepithelial neoplasia and 150 matched health controls to construct the diagnostic model using different cfDNA features. The performance of each diagnostic model was evaluated independently in an external ESCC cohort and a precancerous cohort. Abbreviation: ESCC, esophageal squamous cell carcinoma; IEN, intraepithelial neoplasia; WGS, whole-genome sequencing; WGBS, whole-genome bisulfite sequencing; cfWGBS, cfDNA WGBS; RNAseq, RNA sequencing; HC, healthy control; CNV, copy number variant; DMR, differentially methylated region; IM, immune modulation; CCA, cell cycle pathway activation; IS, immune suppression; NRFA, NRF2 oncogenic activation.

Figure 2. Cell-free DNA methylation markers and their detection performance for esophageal squamous cell carcinoma.

a. Among the differentially methylated regions (DMRs) identified in esophageal squamous cell carcinoma (ESCC) tissues, 650 DMRs were recalled through an adjusted p value < 0.05 (Wilcoxon test), favoring DMRs with ESCC average values more significant than healthy controls in the discovery cohort, as determined by the malignant ratio. The figure shows malignant ratios with the p value of the top ten DMRs as examples. **b.** The diagnostic performances of the ESCC-cfMeth score were evaluated in the discovery cohort (10-fold cross-validation, the curve of each color indicating one cross-validation), the external validation cohort, and the precancerous validation cohort. The black curves represent the receiver operating characteristic (ROC) curves and the blue areas indicate the 95% confidence intervals (CI). **c.** The final prediction model (ESCC-cfMeth

score) was constructed using the cfDNA malignant ratios of the optimal 50 markers. The ESCC-cfMeth scores were significantly higher in patients with ESCC and intraepithelial neoplasia (IEN) than the HCs in the discovery cohort and the validation cohorts. **d.** Compared to the HCs, the ESCC-cfMeth scores were robustly elevated among ESCCs of different stages (left) and IENs (right). **e.** Differential expression of functional genes was found within the 50 optimal DMRs. Abbreviation: ESCC, esophageal squamous cell carcinoma; IEN, intraepithelial neoplasia; LGIEN, low-grade IEN; HGIEN, high-grade IEN; HC, healthy control; DMR, differentially methylated region; AUC, area under curve; CI, confidence interval.

Figure 3. Recalling and analyzing the copy number variant events in cell-free DNA whole-genome bisulfite sequencing data.

a. A whole-genome bisulfite sequencing (WGBS)-based approach for recalling recurrent copy number variants (CNVs) in WGBS data from tissues and cfDNA. **b.** Take Patient 002 in the ECGEA cohort as an example, the amplifications in chr. 3 and chr.5 and deletions in chr. 3, 4, 9, 10, 11, 13, 16,18, and 21 were identified in whole-genome sequencing data and recalled in paired WGBS data. **c.** Compared to the CNV events in healthy controls (HCs), 153 regions had significantly higher CNV event rates in patients with esophageal squamous cell carcinoma (ESCC). The amplifications (red) and deletions (blue) were shown with the corresponding *p* value of the difference in ESCCs vs. HCs. **d.** The CNV-positive rates were significantly higher in patients with ESCC and intraepithelial neoplasia (IEN) than the healthy controls in the discovery cohort and the validation cohorts and positively correlated with the stages and grades. Abbreviation: ESCC, esophageal squamous cell carcinoma; IEN, intraepithelial neoplasia; LGIEN, low-grade IEN; HGIEN, high-grade IEN; HC, healthy control; WGS, whole-genome sequencing; WGBS, whole-genome bisulfite sequencing; GMM, Gaussian Mixture Model; HMM, Hidden Markov Model; CNV, copy number variant; AMPL, amplification; DELE, deletion; G, grade.

Figure 4. Analyzing the cell-free DNA fragment size in whole-genome bisulfite sequencing data.

a. The fragment sizes of cfDNA were surveyed in the discovery cohort. Both the peaks in cfDNA were 166 bp in esophageal squamous cell carcinoma (ESCC) patients and healthy controls. However, more short fragments (90-150bp) were found in the ESCC groups. **b.** The ratio of short fragments in cfDNA was calculated as the fragment size ratio (FSR) in whole-genome bisulfite sequencing data. **c.** The human genome was divided into 5Mb bins, resulting in a total of 1,082 (541 bins \times 2) FSR features. **d.** In the discovery cohort, no significant difference in the average FSR was observed across all bins between ESCC patients and HCs. However, we identified 83 bins where the FSRs were significantly elevated in ESCC patients than HCs in the discovery cohort. The average FSRs in the 83 selected bins were significantly higher in the ESCC patients in the discovery cohort and the external validation cohort, but not the intraepithelial neoplasia patients in the precancerous validation cohort. Abbreviation: ESCC, esophageal squamous cell carcinoma;

IEN, intraepithelial neoplasia; LGIEN, low-grade IEN; HGIEN, high-grade IEN; HC, healthy control; WGS, whole-genome sequencing; WGBS, whole-genome bisulfite sequencing; MAPQ, mapping quality; MDS, multidimensional scaling.

Figure 5. Complementarities of the three cell-free DNA features and the performance of the combined models.

a. The distributions of cfDNA methylation markers, copy number variants, and fragmentation features in the human genome. **b.** Complementarities between these three features were found in the esophageal squamous cell carcinoma (ESCC) patients in the discovery cohort. Each bar indicates the status of each feature in a ESCC patient. **c.** The diagnostic performances of the ESCC-cfMeth score, DMR *plus* CNV model, and EMMA model were evaluated in the external validation cohort and the precancerous validation cohort. **d.** The detection rate of the ESCC-cfMeth score, the DMR *plus* CNV model, and the EMMA model for intraepithelial neoplasia (IEN), stages I, II, and III ESCC. **e.** In the external validation cohort, improved performances of the combined models resulted from the complementarities in three features. **f.** The potential survival benefit of the EMMA model and the ESCC-cfMeth model were estimated according to different test intervals, ranging from 5 years to continuous testing (idealized). **g.** The Schematic diagram of the detection rates for different cfDNA features and combined models and 5-year survival in different stages. Abbreviation: ESCC, esophageal squamous cell carcinoma; IEN, intraepithelial neoplasia; FSR, fragment size ratio; CNV, copy number variant; DMR, differentially methylated region; EMMA, expanded multimodal analysis; Meth, methylation; hypo, hypomethylation; hyper, hypermethylation; AUC, area under curve; CI, confidence interval.

Figure 6. The biological significance of the DNA methylation markers in esophageal squamous cell carcinoma cell-free DNA.

a. We divided the 155 patients with esophageal squamous cell carcinoma (ESCC) in the ECGEA cohort into three groups according to the average methylation level of 50 optimal DNA methylation markers in the ESCC-cfMeth score. **b.** The proportions of molecular subtypes of ESCC in the methylation-dominant, methylation-moderate, and methylation-poor groups. **c.** The cell components in the tumor microenvironment were estimated between the methylation-dominant group and the methylation-moderate/poor groups. **d.** The pathways were enriched in the methylation-dominant group and the methylation-poor group. Abbreviation: CNV, copy number variant; COCA, cluster of cluster assignments; CIMP, CpG island methylator phenotype; Meth-cluster, methylation cluster; DMR, differentially methylated region; hypo, hypomethylation; hyper, hypermethylation; CCA, cell cycle pathway activation; IM, immune modulation; IS, immune suppression; NRFA, NRF2 oncogenic activation; DC, dendritic cell.

Extended Data Figure 1. Plasma cfDNA concentrations, sequencing coverage, and selection

of the optimal number of cfDNA methylation markers.

a. No significant difference was found in the cfDNA concentrations between the patients with esophageal squamous cell carcinoma (ESCC) and healthy controls (HCs). **b.** The whole-genome bisulfite sequencing (WGBS) data of cfDNA covered 89% of the reference genome on average with $9.51\times$ depth. **c.** Using data from the discovery cohort, a random forest algorithm was adopted to generate prediction models using the cfDNA malignant ratios of the top 1 to 650 differentially methylated regions (DMRs). The top 50 DMRs achieved the optimal performance for distinguishing between malignant and benign plasma samples. Abbreviation: ESCC, esophageal squamous cell carcinoma; IEN, intraepithelial neoplasia; HC, healthy control.

Extended Data Figure 2. Cell-free DNA methylation markers selection.

Among the 650 differentially methylated regions (DMRs) of esophageal squamous cell carcinoma (ESCC) cfDNA in the discovery cohort, the optimal 50 DMRs included 40 hypo-DMRs and 10 hyper-DMRs. Abbreviation: DMR, differentially methylated region; hypo, hypomethylation; hyper, hypermethylation.

Extended Data Figure 3. The malignant ratios of the optimal differentially methylated regions and the potential biological significance of the functional genes within them in early-stage ESCC.

a. The 50 optimal differentially methylated regions (DMRs) in the ESCC-cfMeth model. The malignant ratios in these regions were significantly different between the ESCC patients and healthy controls in the discovery cohort. * $p \leq 0.05$; ** $p \leq 0.01$; *** $p \leq 0.001$. **b.** To reveal the biological significance of the functional genes within the 50 optimal DMRs in early-stage ESCC, we analyzed the expression levels of these genes in 10-pair tissue samples of stage-I ESCC and normal tissues from a published dataset [GSE213565]. *ZNF132* with a hypermethylated promoter displayed significant down-regulation, and *LINC00680* with hypomethylation within its gene body showed upregulation. In addition, although there is no statistical significance, *FLT1* and *ID1* were also up-regulated. Abbreviation: ESCC, esophageal squamous cell carcinoma; HC, healthy control; TPM, transcripts per million.

Extended Data Figure 4. Performances of short fragment ratios in 83 bins.

The diagnostic performances of the fragment size ratios (FSRs) in the 83 selected regions where FSRs were significantly elevated in ESCC patients than HCs in the discovery cohort were evaluated in the discovery cohort (10-fold cross-validation), the external validation cohort, and the precancerous validation cohort. Abbreviation: ESCC, esophageal squamous cell carcinoma; HC, healthy control; AUC, area under curve; CI, confidence interval.

Extended Data Figure 5. Complementarities of the three cfDNA features.

a. The overlapping of the cfDNA methylation markers (differentially methylated regions, DMRs), copy number variants, and fragmentation features in the human genome. **b.** There was no significant association between the ESCC-cfMeth score, average fragment size ratio, and the copy number variant (CNV) events in the discovery cohort. **c.** The diagnostic performances of the DMR

plus CNV and EMMA models were evaluated in the discovery cohort (10-fold cross-validation). **d.** In the precancerous validation cohort, improved performances of the combined models resulted from the complementarities in three features. Notably, The EMMA model detected additional patients with intraepithelial neoplasia which were negative in all three single-modal models. Abbreviation: DMR, differentially methylated region; FSR, fragment size ratio; CNV, copy number variant; AUC, area under curve; CI, confidence interval; IEN, intraepithelial neoplasia; LGIEN, low-grade IEN; HGIEN, high-grade IEN; EMMA, expanded multimodal analysis.

Extended Data Figure 6. The specificity of the DNA methylation markers in the ESCC-cfMeth model across cell types.

The methylation level of the 50 differentially methylated regions of the ESCC-cfMeth model in 81 common cell types³⁷ was analyzed and compared to the primary tumors of ESCC and matched adjacent nonneoplastic tissues from the ECGEA cohort²⁰. Abbreviation: ESCC, esophageal squamous cell carcinoma; DMR, differentially methylated region; hypo, hypomethylation; hyper, hypermethylation; ECGEA, the ESCC Genome and Epigenome Atlas.

Extended Data Figure 7. The overall survival of the ESCC patients in methylation-dominant group and the methylation-moderate/poor groups.

The survival of the methylation-dominant group and that of the rest two groups were compared in all grades and grades I-III. Abbreviation: ESCC, esophageal squamous cell carcinoma.

Extended Data Figure 8. The molecular and clinical characteristics of the methylation-dominant, methylation-moderate, and methylation-poor groups.

The proportions of positive CpG island methylator phenotype (CIMP), copy number variant (CNV), gender, stage, grade, location, smoking, drinking, status of *TP53* and APOBEC genes, and the APOBEC mutational signatures in the methylation-dominant, methylation-moderate, and methylation-poor groups. The proportion of the APOBEC mutational signatures was also compared between the ESCC patients with or without CNVs. Abbreviation: CIMP, CpG island methylator phenotype; CNV, copy number variant; MD, methylation-dominant group; MM, methylation-moderate group; MP, methylation-poor group; NS, not significant.

Reviewer #2 (Remarks to the Author): expertise in DNA methylation bioinformatics

In the manuscript entitled “Expanded multimodal analysis of cell-free DNA methylomes for ultra-early detection of esophageal squamous cell carcinoma and precancerous lesions' ". Firstly, the authors showed the functional relevance of DNA methylation, Copy number variations (CNV) and short fragments ratio (fragment size ratio, FSR) in cfDNA with ESCC and precancerous lesions. Furthermore, they developed a machine-learning based modal called EMMA, simultaneously considering differential DNA methylation regions (DMR), CNV and FSR information from the whole genome bisulfite sequencing data of cell-free DNA from 230 non-metastatic esophagus squamous cell carcinoma or precancerous lesions patients' plasma samples and 230 matched healthy control plasma samples. The authors claimed that EMMA has both higher sensitivity and specificity than those using only DMR, DMR combined with CNV data. At last, they further classified the 155 patients samples into 3 groups based on cfDNA methylation status hyper, moderate and hypo methylation group. And found that the cfDNA methylation groups correlates with the subtypes, cell type enrichment and gene expression activations of the ESCC tissues, further suggesting the potential of cfDNA methylation as an important factor for ESCC subtyping and treatment guidance. Below are some questions that need to be clarified.

Response: Thank you for the remarks and insightful synopsis of our study. Please see our point-to-point responses to specific comments below.

1. It's a large patient sample cohort in the manuscript. Have the patients undergone any medical treatments before they do blood draw or tissue sampling?

Response: Thank you for your professional comment. The blood and tissue were obtained from each participant before any medical intervention. We highlighted this description in the revised manuscript as the following.

“Our study encompassed a diverse cohort, including 150 **untreated** patients with ESCC or esophageal high-grade intraepithelial neoplasia (HGIEN, namely stage-0 ESCC) from the Cancer Hospital of the Chinese Academy of Medical Sciences and Peking Union Medical College in Beijing, China (CHCAMS, the discovery/training cohort), 30 **untreated** ESCC patients from the Shanghai Chest Hospital in Shanghai, China (the external validation cohort), 50 patients with esophageal IEN from CHCAMS (the precancerous validation cohort), and 230 HCs, each age- and gender-matched within their respective cohorts (Fig. 1b; Extended Data Table 1). We collected a median of 2 mL of plasma from each participant (n=460) before any medical intervention.” (Page 7, Lines 12-20)

“The ECGEA cohort consisted of 155 **untreated** patients diagnosed with ESCC between May 2017 and July 2018 at Shanxi Cancer Hospital, China. Samples of ESCC tissue and paired adjacent normal/non-neoplastic tissue were collected before **any** treatment.” (Page 22, Lines 10-12)

2. Could you make detailed explanations for some figures? For example. In Fig.2a, what is the difference between the left and right panel?

Response: Thank you for your constructive suggestion. We apologize for the misleading in Fig. 2a. There was no difference between the left and right panels. Thus, we've merged these two panels

in the revised Fig. 2a. We've rewritten all the figure legends in the revised manuscript including that of Fig. 2a as follows.

Revised Figure 2. Cell-free DNA methylation markers and their detection performance for esophageal squamous cell carcinoma.

a. Among the differentially methylated regions (DMRs) identified in esophageal squamous cell carcinoma (ESCC) tissues, 650 DMRs were recalled through an adjusted p value < 0.05 (Wilcoxon test), favoring DMRs with ESCC average values more significant than healthy controls in the discovery cohort, as determined by the malignant ratio. The figure shows malignant ratios with the p value of the top ten DMRs as examples.

3. In Fig.2e, the authors showed the differential expression of a few classical genes between ESCC and HC samples. How about those in the precancerous cohort?

Response: Thank you for your professional comment. Unfortunately, there is no sufficient tissue sample in our precancerous cohort or available published data for esophageal squamous precancerous lesions (ESPL). According to a recently published study, this might result from that: 1) fresh ESPL tissues obtained from biopsies under the endoscopy are not sufficient for RNA sequencing and 2) the majority of the tissue sample is composed of normal epithelia (Liu X, *et al. Nat Commun.* 2023;14[1]:4779; PMID: 37553345). We've added this limitation in the revised manuscript as “the potential function of these multimodal markers is still unknown, especially in precancerous lesions” (Page 20, Lines 16-17).

To reveal the biological significance of these classical functional genes in early-stage esophageal squamous cell carcinoma (ESCC), we analyzed and validated the expression levels of these genes in 10-pair tissue samples of stage-I ESCC and normal tissues from a published dataset [GSE213565, <https://www.ncbi.nlm.nih.gov/geo/query/acc.cgi?acc=GSE213565>] (Zhao J, *et al. BMC Med Genomics.* 2023;16[1]:153; PMID: 37393256). Similar to the result from our ECGEA cohort (the ESCC Genome and Epigenome Atlas; Liu, Z. *et al. Cancer Cell.* 2023;41:181-195.e189; PMID: 36584672), *ZNF132* with a hypermethylated promoter displayed significant down-regulation, and *LINC00680* with hypomethylation within its gene body showed upregulation. It indicates their potential function in early-stage ESCC. In addition, although there is no statistical significance, *FLT1* and *ID1* were also up-regulated. We've added this result in the **Result** section and **Extended Data Figure 3b** in the revised manuscript as follows.

“To reveal the biological significance of these classical functional genes in early-stage ESCC,

we analyzed and validated the expression levels of these genes in 10-pair tissue samples of stage-I ESCC and normal tissues from a published dataset [GSE213565]²⁸. Similarly, *ZNF132* was significantly down-regulated and *LINC00680* was significantly up-regulated, which indicates their potential function in early-stage ESCC. However, *FLT1* and *ID1* were also up-regulated but without significance in stage-I ESCC (Extended Data Figure 3b).” (Page 9, Lines 13-19)

Reference in this section:

28. Zhao, J. et al. Genomic and transcriptional characterization of early esophageal squamous cell carcinoma. *BMC Med Genomics* 16, 153 (2023). <https://doi.org/10.1186/s12920-023-01588-7>

Extended Data Figure 3. The malignant ratios of the optimal differentially methylated regions and the potential biological significance of the functional genes within them in early-stage ESCC.

b. To reveal the biological significance of the functional genes within the 50 optimal DMRs in early-stage ESCC, we analyzed the expression levels of these genes in 10-pair tissue samples of stage-I ESCC and normal tissues from a published dataset [GSE213565]. *ZNF132* with a hypermethylated promoter displayed significant down-regulation, and *LINC00680* with hypomethylation within its gene body showed upregulation. In addition, although there is no statistical significance, *FLT1* and *ID1* were also up-regulated.

Abbreviation: ESCC, esophageal squamous cell carcinoma; HC, healthy control; TPM, transcripts per million.

4. Gene mutations such as TP53 and APOBEC was frequently found in ESCC patients (PMID: 36266286, PMID: 34663923,). What is the gene mutations/ SNV status in cfDNA DMR, CNV and FSR regions? Could you check if the TP53 and APOBEC mutations are in the DMR?

Response: Thank you for your professional suggestions. According to your suggestion, we check the overlapping of *TP53* and *APOBEC* genes with the multimodal markers including the cfDNA methylation markers (differentially methylated regions, DMRs), copy number variants (CNVs), and fragmentation features (fragment size ratios, FSRs). As a result, we found these genes were not within the regions of either DMRs or CNVs, while *TP53* and *APOBEC3* genes were within the regions of FSRs. We’ve added these results in the **Result** section and **Extended Data Table 6**

to the revised manuscript as shown below.

“Mutations in *TP53*³⁸ and APOBEC genes³⁹ as well as the APOBEC mutational signatures (SBS2/13)⁴⁰ were frequently observed in ESCC patients... We also investigated the overlapping of *TP53* and APOBEC genes with the multimodal markers and found these genes were not within the regions of either DMRs or CNVs (Extended Data Table 6). However, *TP53* and APOBEC3 genes were within the regions of FSRs, which indicates a probably higher proportion of ctDNA in cfDNA in these regions. Thus, identifying the mutations in genes like *TP53* in cfDNA might further improve the detection rate of ESCC.” (Page 17 Lines 2-12)

References in this section:

38. Murai, K. et al. p53 mutation in normal esophagus promotes multiple stages of carcinogenesis but is constrained by clonal competition. *Nat Commun* 13, 6206 (2022). <https://doi.org/10.1038/s41467-022-33945-y>
39. Moody, S. et al. Mutational signatures in esophageal squamous cell carcinoma from eight countries with varying incidence. *Nat Genet* 53, 1553-1563 (2021). <https://doi.org/10.1038/s41588-021-00928-6>
40. Wang, Y. et al. APOBEC mutagenesis is a common process in normal human small intestine. *Nat Genet* 55, 246-254 (2023). <https://doi.org/10.1038/s41588-022-01296-5>

Extended Data Table 6. The overlapping of the *TP53* and APOBEC genes and the ESCC cfDNA marker

Gene	Chr	Start	End	Candidate markers
TP53	chr17	7661779	7687550	FSR chr17:5000000-10000000
The APOBEC Family				
AICDA	chr12	8602170	8612867	None
APOBEC1	chr12	7649400	7665908	
APOBEC2	chr6	41053304	41064511	
APOBEC4	chr1	183646275	183653316	
APOBEC3A	chr22	38952741	38992778	FSR chr22:35000000-40000000
APOBEC3B	chr22	38982347	38992804	
APOBEC3C	chr22	39014257	39020352	
APOBEC3D	chr22	39021113	39033277	
APOBEC3F	chr22	39040604	39055972	
APOBEC3G	chr22	39077067	39087743	
APOBEC3H	chr22	39097224	39104067	

Abbreviation: APOBEC, apolipoprotein B mRNA-editing enzyme catalytic polypeptide-like; ESCC, esophageal squamous cell carcinoma; cfDNA, cell-free DNA; chr, chromosome; FSR, fragment size ratio.

However, the mutation calling has not been conducted in the whole-genome bisulfite

sequencing (WGBS) data due to the two following concerns. First, as the bisulfite treatment used in WGBS converts unmethylated cytosines/C to uracils/U (interpreted as thymines/T during subsequent sequencing), the conversion seriously interferes with the original DNA mutation information regarding unmethylated cytosine positions. Second, the sequencing depth and coverage of the WGBS data generated from the cfDNA is insufficient to detect low-frequency DNA mutations accurately. Thus, we believe the current mutation calling method in WGBS is not reliable for sufficient accuracy. However, the overlapping of *TP53* and *APOBEC3* genes and the regions of FSRs might indicate a probably higher proportion of ctDNA in cfDNA in these regions. Thus, identifying the mutations in genes like *TP53* in cfDNA might further improve the detection rate of ESCC.

5. Could you add the WGS and RNA-seq library preparation for the tissues in the method part?

Response: Thank you for your suggestion. We apologize for not describing this part of the method. As these data were generated from our ECGEA (the ESCC Genome and Epigenome Atlas) cohort, we've provided a detailed description of these related methods in our previous study (Liu, Z. *et al. Cancer Cell*. 2023;41:181-195.e189; PMID: 36584672). We've added this explanation to our revised manuscript as follows.

“Whole-genome multi-omic analysis in ESCC tissues and paired adjacent normal tissue

Genomic DNA and total RNA were extracted from the primary tumors and matched adjacent non-neoplastic tissues of 155 ESCC cases from our previous ECGEA cohort. A total of ~300 ng of high-quality DNA was used for constructing the WGS library and sequencing, while ~200 ng of high-quality DNA was used for WGBS library preparation and sequencing. About 1 µg of high/medium-quality RNA was used for library construction and sequencing. The detailed method and data have been described in our previous study.” (Page 23 Lines 2-8)

6. In Fig. 5c, the AUC of EMMA model didn't outperform that much compared with that using the DMR plus CNV model. Have you tried to change the short fragments size range other than 90 to 150bp for the EMMA model? If combined with *TP53* and *APOBEC* mutation status from the cfDNA, what is the performance of EMMA look like?

Response: Thank you for your professional suggestion. Indeed, there was no statistical difference between the performances in the EMMA model and the DMR *plus* CNV model. It might be due to the cfDNA fragments damaging during bisulfite conversion in WGBS weakening the signals of fragmentation features. However, a significant difference was only observed between the EMMA model and the ESCC-cfDNA model, but not between the DMR *plus* CNV model and the ESCC-cfDNA model, indicating the potential contribution of the FSRs ($p=0.045$ and 0.8075). The short fragment size range from 90 to 150bp has long been demonstrated to lead to two-fold enrichment of ctDNA in 95% of tumor patients (Mouliere F, *et al. Sci Transl Med*. 2018;10:eaat4921; PMID: 30404863). Thus, this fragment range is widely adopted in most studies on cfDNA fragmentomics in cancers (Peneder P, *et al. Nat Commun*. 2021;12[1]:3230; PMID: 34050156. Christodoulou E, *et al. NPJ Precis Oncol*. 2023;7[1]:21; PMID: 36805676). This phenomenon reflects the rapid proliferation characteristics of cancer cells, their higher mortality rate, and an abnormal pattern of

chromatin accessibility (Markus H, *et al. Sci Transl Med.* 2021;13[581]; PMID: 33597261. Lubotzky A, *et al. Elife.* 2022;11; PMID: 35699419). We also validated it in our data and found the differential frequencies in fragment size were more obvious in the range from 90 to 150bp between the ESCC cases and healthy controls (**Figure R2**). Thus, cfDNA fragments of 90-150 bp were determined as short fragments in further analysis.

Figure R2. Higher frequencies of short cfDNA fragments (90-150bp) in the ESCC cases.

Furthermore, we investigated the association between the somatic mutation rates in *TP53* and APOBEC genes and the proportion of the APOBEC mutational signatures with the three methylation subgroups and CNV events. However, there were no significant differences in the somatic mutation rates in *TP53* and APOBEC genes and the proportion of the APOBEC mutational signatures between the three groups (all $p > 0.05$). In ESCC, the significance of APOBEC mutational signatures underscores their crucial role in tumor progression and prognosis, particularly in advanced stages, lymph node metastasis, and decreased overall survival rates in patients (Cui Y, *et al. Cell Res.* 2020;30[10]:902-913; PMID: 32398863). Subsequently, we examined the proportion of the APOBEC mutational signatures, which was found to have no association with the presence of copy number variations (CNV) in ESCC patients. We've added these results to the **Result** section in the revised manuscript and the revised **Extended Data Figure 8** as shown below.

“Mutations in *TP53*³⁵ and APOBEC genes³⁶ as well as the APOBEC mutational signatures (SBS2/13)³⁷ were frequently observed in ESCC patients. However, there were no significant differences in the somatic mutation rates in *TP53* and APOBEC genes and the proportion of the APOBEC mutational signatures between the three groups (all $p > 0.05$; Extended Data Figure 8). The proportion of the APOBEC mutational signatures was also not associated with the status of carrying CNV in ESCC patients... Thus, identifying the mutations in genes like *TP53* in cfDNA might further improve the detection rate of ESCC.” (Page 17 Lines 2-12)

Extended Data Figure 8. The molecular and clinical characteristics of the methylation-dominant, methylation-moderate, and methylation-poor groups.

The proportions of positive CpG island methylator phenotype (CIMP), copy number variant (CNV), gender, stage, grade, location, smoking, drinking, **status of *TP53* and APOBEC genes, and the APOBEC mutational signatures** in the methylation-dominant, methylation-moderate, and methylation-poor groups. **The proportion of the APOBEC mutational signatures was also compared between the ESCC patients with or without CNVs.**

Thus, the mutation status of *TP53* was not associated with the DNA methylation signatures or the CNV events. Theoretically, accurate identification of the somatic mutations in cfDNA using high-depth next-generation sequencing or droplet digital PCR could further detect 70% missing ESCC cases from EMMA or other cfDNA methylation-based approaches according to its high mutation rate (>70%). We added this description to the limitation in the revised manuscript as follows.

“Fourth, as the mutation calling is unreliable in WGBS data, mutations were not enrolled in this multimodal model. While accurately identifying the mutations in genes like *TP53* in cfDNA using additional approaches might further improve the detection rate of ESCC.” (Page 20 Lines 21-23)

7. The data in the manuscript can't fully support the title that the EMMA model enables ultra-early detection of ESCC. For example, from Fig.5d, the EMMA model's performance becomes better with the progress of the tumor stages. Please consider changing “ultra early” to a moderate description of the modal.

Response: Thank you very much for your suggestion. Indeed, the detection abilities of the EMMA model increased with the progress of the ESCC stages. We've changed “ultra-early” to “early” as a moderate description and added the term “molecular subtyping” to comprehensively describe our study. Thus, the revised title is “**Expanded multimodal analysis of cell-free DNA methylomes for early detection and molecular subtyping of esophageal squamous cell carcinoma and precancerous lesions**”.

Reviewer #3 (Remarks to the Author): expertise in ESCC clinical biomarkers

Understanding the combined performances and biological significance of different cfDNA features is important to exploring non-invasive methods for cancer early detection. In this study, Liu et al. reported a novel computational framework for the early diagnosis of ESCC through multi-modal analysis of whole-genome methylation data from cfDNA. The authors systematically revealed the timing and complementarity of different cfDNA features and obtained high detection rates of precancerous lesions and early-stage disease. Moreover, the diagnostic DNA methylation features were found to be associated with distinct molecular subtypes and tumor microenvironments. This study also provides a high-quality dataset of cfDNA methylome, which would further benefit future studies on liquid biopsy of ESCC. Overall, this is a solid study with a meticulous design, stringent analysis, and novel findings. However, there are still some minor concerns that need to be clarified.

Response: Thank you very much for your constructive comment and the affirmation of the manuscript regarding our study: “Overall, this is a solid study with a meticulous design, stringent analysis, and novel findings”.

Please see our point-to-point responses below.

1. To estimate the potential benefits to 5-year overall survival rates of ESCC by the multi-modal cfDNA model, the authors conducted an adapted interception analysis. The detailed baseline data used in the analysis and their sources should be provided in the method.

Response: Thank you for your professional suggestion. We’ve added this description in the **Method** section (“*Clinical benefits estimation*”) in the revised manuscript as follows.

“The clinical benefits were estimated according to current stage-specific diagnostic yields of ESCC³⁵, the average annual rate of progression from IEN to ESCC^{6,36}, the shift rates of stages in ESCC³⁴, and the 5-year overall survival rates of IENs⁶ and ESCCs of different stages³⁵.” (Pages 28-29)

References in this section:

6. Zhang, Y. Q. et al. Endoscopic Submucosal Dissection for Superficial Proximal Esophageal Neoplasia is Highly Successful. *Ann Surg* 266, 995-999 (2017).
34. Hubbell, E., Clarke, C. A., Aravanis, A. M. & Berg, C. D. Modeled Reductions in Late-stage Cancer with a Multi-Cancer Early Detection Test. *Cancer Epidemiol Biomarkers Prev* 30, 460-468 (2021). <https://doi.org:10.1158/1055-9965.Epi-20-1134>
35. He, Y. et al. Clinical characteristics and survival of 5283 esophageal cancer patients: A multicenter study from eighteen hospitals across six regions in China. *Cancer Commun (Lond)* 40, 531-544 (2020). <https://doi.org:10.1002/cac2.12087>
36. Wei, W. Q. et al. Esophageal Histological Precursor Lesions and Subsequent 8.5-Year Cancer Risk in a Population-Based Prospective Study in China. *Am J Gastroenterol* 115, 1036-1044 (2020).

2. To demonstrate the advantages of the multi-modal approach, the potential survival benefits should be compared between the EMMA model and the cfDNA-cfMeth model in at least one assumed clinical scenario.

Response: Thank you very much for your constructive comment. According to your suggestion, we've compared the survival benefits between the EMMA model and the ESCC-cfMeth model. We also revised **Figure 5f** and added the comparison to the **Result** section in the revised manuscript as follows.

“When the ESCC-cfMeth and EMMA approaches are implemented, the 5-year overall survival rates of ESCC patients in China show potential improvement by early detection and intervention at the IEN stage (**Fig. 5f**). Comparatively, the EMMA model, which integrates multi-modal data, demonstrates higher survival benefits in the assumed clinical scenario compared to the ESCC-cfMeth model, which solely utilizes cfDNA methylation. Depending on the test interval, ranging from 5 years to continuous testing (idealized), the ESCC-cfMeth model exhibited a potential increase in 5-year overall survival rates by 26.90% to 35.25%. In contrast, the EMMA model shows a potential increase of 33.87% to 41.95%. Specifically, when the EMMA approach is applied annually at the IEN stage, the 5-year overall survival of ESCC patients in China could further increase by 6.9% compared to the ESCC-cfMeth model (40.22% vs. 33.37%).” (Page 14 Lines 9-19)

Revised Figure 5. Complementarities of the three cell-free DNA features and evaluating the performance of the combined models.

f. The potential survival benefit of the EMMA model and the ESCC-cfMeth model were estimated according to different test intervals, ranging from 5 years to continuous testing (idealized).

3. As the authors stated in the limitation, the external validation cohort is relatively small. How was the sample size for the external validation cohort determined?

Response: Thank you for your professional comment. As there is no pre-specified sample size for the discovery cohort was determined, we enrolled the patients with esophageal squamous cell carcinoma (ESCC) or esophageal high-grade intraepithelial neoplasia (HGIEN, namely stage-0 ESCC) and gender- and age-matched healthy controls (HCs) at the Cancer Hospital of the Chinese Academy of Medical Sciences and Peking Union Medical College (CHCAMS) between May 2019 and December 2022. Furthermore, we determined the sample size of the validation cohorts using PASS software. The parameters were set as $\alpha=0.05$, $1-\beta=0.90$, $AUC=0.90$ (10-fold cross-validation in the discovery cohort), and null hypothesis $AUC=0.70$. The results showed that the number of positive and negative cases was 30. Thus, we enrolled 30 patients with ESCC and 30 matched HCs from the Shanghai Chest Hospital as the external validation cohort. Additionally, to ensure the generalizability of our findings to patients with precancerous lesions, we included a precancerous validation cohort consisting of 50 patients with IEN and 50 matched healthy controls from the CHCAMS.

4. The cut-off value of 0.5 for the ESCC-cfMeth score was used to differentiate between ESCC and normal samples. How was this value set?

Response: Thank you for your comment. We developed the ESCC-cfMeth score using a random forest model with default settings of 500 decision trees and randomized variables being chosen at each split. This score predicts the likelihood of a sample having ESCC, ranging from 0/0% to 1/100%. To distinguish between ESCC samples and non-ESCC/healthy samples, we set a cut-off value of 0.5, namely the possibility of 50%. Thus, samples with an ESCC-cfMeth score $\geq 0.5/50\%$ were classified as ESCC samples, while those with a score below 0.5/50% were considered non-ESCC/healthy.

5. Both the abbreviations “EIN” and “IEN” are used in the manuscript and figures, which is unclear.

Response: Thank you very much for your comment. We apologize for these inconsistent abbreviations. We have unified them into “IEN” in the entire manuscript and the legends of Figures 2-4 and Extended Data Figure 4.

6. The web links for the data and code generated in this study should be provided in the “Availability of data and materials”.

Response: Thank you for your suggestion. All the raw data generated in this study have been deposited in the Genome Sequence Archive (GSA) for the Human database. The code was provided in an open-source repository in Git Hub. We’ve added a detailed description in the “Availability of data and materials” as follows.

“The whole-genome bisulfite sequencing (WGBS) data from 460 cfDNA samples generated in this study have been deposited in the Genome Sequence Archive (GSA) for Human database in the BIG Data Center (<http://bigd.big.ac.cn/gsa>), Beijing Institute of Genomics (BIG), Chinese Academy of Sciences with accession number **HRA006113** (<https://ngdc.cncb.ac.cn/gsa->

human/browse/HRA006113). According to the guidelines of GSA-human, all non-profit researchers and the principal investigators of any research group are allowed access to the data. Request for this data access should be addressed to the corresponding author Zhihua Liu (liuzh@cicams.ac.cn). The multi-omics genome-wide data from tissue samples is available through the GSA database for Human database under accession code HRA003107 (whole-genome sequencing & RNA-seq, <https://ngdc.cncb.ac.cn/gsa-human/browse/HRA003107>) and HRA003533 (WGBS, <https://ngdc.cncb.ac.cn/gsa-human/browse/HRA003533>). The code was provided in an open source repository in [github](https://github.com/packageandcode/EMMA) (<https://github.com/packageandcode/EMMA>).” (Page 31 Lines 11-22)

REVIEWERS' COMMENTS

Reviewer #1 (Remarks to the Author):

the authors have responded very well to my comments, and the comments of other reviewers, and clarified many of the comments in detail, with additional revisions, to improve the quality of the work.

Reviewer #2 (Remarks to the Author):

Thank you for the comprehensive answers. The manuscript looks good to me now.

Reviewer #3 (Remarks to the Author):

The authors have solved all my concerns and I recommend the manuscript to be accepted.

Reviewer #3 (Remarks on code availability):

The code is a usable resource for researchers.